

# Subsystem symmetry fractionalization and foliated field theory

**Po-Shen Hsin[1], David T. Stephen[2,3], Arpit Dua[3] and Dominic J. Williamson[4∘]**

**1** Mani L. Bhaumik Institute for Theoretical Physics,
475 Portola Plaza, Los Angeles, CA 90095, USA
**2** Department of Physics and Center for Theory of Quantum Matter,
University of Colorado, Boulder, CO 80309, USA
**3** Department of Physics and Institute for Quantum Information and Matter,
Caltech, Pasadena, CA, USA
**4** Centre for Engineered Quantum Systems, School of Physics,
University of Sydney, Sydney, NSW 2006, Australia

## Abstract

Topological quantum matter exhibits a range of exotic phenomena when enriched by subdimensional symmetries. This includes new features beyond those that appear in the conventional setting of global symmetry enrichment. A recently discovered example is a type of subsystem symmetry fractionalization that occurs through a different mechanism to global symmetry fractionalization. In this work we extend the study of subsystem symmetry fractionalization through new examples derived from the general principle of embedding subsystem symmetry into higher-form symmetry. This leads to new types of symmetry fractionalization that are described by foliation dependent higher-form symmetries. This leads to field theories and lattice models that support previously unseen anomalous subsystem symmetry fractionalization. Our work expands the range of exotic topological physics that is enabled by subsystem symmetry in field theory and on the lattice.



## Contents

---

∘ DW's current address is: IBM Quantum, IBM Almaden Research Center, San Jose, CA 95120, USA.

# 1 Introduction

Global symmetries play an important role in understanding universality classes of quantum many-body systems via the Landau-Ginzburg paradigm of phase transitions [1]. In this paradigm, it is crucial to understand how relevant objects in the system of interest transform under the global symmetry. In a conventional system, the local operators can be divided according to the linear representation they carry under a standard global symmetry. On the other hand, extended operators such as line or surface operators can also transform non-trivially under a conventional symmetry. This occurs when the symmetry has an anomaly on the region supporting the extended operators (see *e.g.* Refs. [2, 3]). The representation carried by an

extended object in this generalized setting is referred to as symmetry fractionalization [4–6]. Symmetry fractionalization has a rich history dating back to the fractional quantum Hall effect, where it is essential in understanding that quasiparticle line operators carry a projective representation of the global symmetry and contribute to the Hall conductance [7, 8].

Recently, there have been significant developments in the understanding of symmetry fractionalization in 2+1D gapped systems using symmetry defects [4–6]. This was extended to 3+1D in a series of works [9–13]. Later it was also understood that fractionalization can be described in terms of higher symmetry [14–16] (see *e.g.* Ref. [17] for a recent review) in gapped or gapless quantum systems in any spacetime dimension. Fractionalization of ordinary symmetry is described in Ref. [2, 3, 18–22] as embedding the ordinary symmetry into higher symmetries that act naturally on extended operators. For instance, a conventional global $U(1)$ symmetry can be embedded into one-form symmetry by specifying the relation of their background gauge field:

$$B_2 = \alpha dA, \tag{1}$$

where $B_2$ is the two-form gauge field for the one-form symmetry, $A$ is the one-form gauge field for the conventional $U(1)$ global symmetry, and $\alpha \in \mathbb{R}/(2\pi\mathbb{Z})$ is a parameter of the embedding. This means that the line operator that transforms under the one-form symmetry with charge $q^{(1)}$ is attached to the Wilson surface $q^{(1)} \int_\Sigma B_2 = \alpha q^{(1)} \int_\Sigma dA = \alpha q^{(1)} \int_{\partial\Sigma} A$, and the operator carries fractional charge $\alpha q^{(1)}$ of the conventional $U(1)$ symmetry. The aforementioned method also applies to fractionalization of higher symmetry such as in Ref. [3, 18]. More recently, there have been developments of higher representation theory for fractionalization of conventional global symmetries on line and surface operators, see *e.g.* Ref. [23, 24].

In this work, we develop field theoretic methods to study the fractionalization of subsystem symmetry, *i.e.* symmetry that only acts on rigid lower-dimensional subsets of a lattice many-body system. Such fractionalization phenomena have recently been discovered in a variety of lattice models [25, 26] following earlier work on subsystem symmetry and topological order [27–36]. We focus on systems with fully mobile excitations, and subsystem symmetries that are derived from subgroups of one-form symmetries. Examples of such subsystem symmetries include operators supported on lines in two spatial dimensions and planes in three spatial dimensions. The fields that describe subsystem symmetry are foliated gauge fields [37–39], which we briefly review in Appendix A (foliated field theories are also related to higher-rank tensor gauge theory, see *e.g.* Ref. [40–43] and the references therein).

We use the following terminology throughout this work: a global symmetry, as characterized by a generator that commutes with the Hamiltonian, is called a *q-form* symmetry whenever the generator has support on a codimension-$q$ subspace in space [14]. A symmetry is called a *subsystem* symmetry if the symmetry generator is not fully topological, i.e. the eigenvalue of the generator changes if we deform its support in general directions, even when it is away from other operators. These adjectives apply to global symmetries, and when a symmetry obeys the above two properties, we call it a *subsystem q-form* symmetry.

A subsystem one-form symmetry is described by a foliated two-form gauge field $B_2^k$, where $k$ labels the foliation, that satisfies the condition

$$B_2^k e^k = 0, \tag{2}$$

for the foliation one-form $e^k$. For instance, if the foliation one-form is $e^k = dx$, then the condition means that $B_2^k$ only has $dxdt, dxdy$ components for the other spatial coordinate $y$ and time coordinate $t$. This describes one-form symmetries whose generator in any spatial slice is supported on codimension-one (with respect to space) submanifolds of constant $x$ coordinate. If we take space to be the $xy$-plane, then the subsystem one-form symmetry is supported on a straight lines in the $y$ direction. The fractionalization of the one-form symmetry

can be described by embedding it into the entire one-form symmetry:

$$B_2 = \sum_k v^k B_2^k,\tag{3}$$

where $v^k$ are integers, and we sum over different foliations $k$. This raises two immediate questions, which we address in this work:

(1) what kind of fractionalization does the above embedding relation describe?

(2) How does one translate the fractionalization from a field theory to a lattice model?

Question (2) requires understanding the topological nature of higher-form symmetry generators in lattice models, while (1) is related to the foliation independence of symmetry generators.

## 1.1 Introduction to the main ideas

Here we summarize the main ideas on constructing subsystem symmetries from higher-form symmetries and relating foliation dependence to subsystem symmetry fractionalization.

### 1.1.1 Higher-form symmetry and its topological nature

Systems with fully mobile excitations often possess $n$-form higher symmetries that act on the extended operators that create these excitations [14]. The background gauge fields that describe the bundle for $n$-form symmetry are anti-symmetric $(n+1)$-form gauge fields [14]. The $n$-form higher symmetry can either be *fully topological*, i.e. invariant under continuous deformations,[1] or only *partially topological*, i.e. only deformable when applied to ground states of the system. The first case arises in field theory, while the latter case is typical in lattice Hamiltonian models. For instance, Kitaev's toric code model in 2+1D has one-form symmetry supported on closed loops that commute with the Hamiltonian. Such one-form symmetry is not fully topological, because a small contractible loop acts non-trivially on the entire Hilbert space of excited states. On the other hand, if we restrict to the space of ground states, a small contractible loop acts trivially, and hence the symmetry becomes topological only on the ground states subspace. Such a distinction is important in our discussion below.

**Fully topological symmetry from gauging contractible symmetry.** Let us begin with a model where the higher-form symmetry is only topological on the states without higher-form charge. By projecting out such states, we can obtain a new model where the higher-form symmetry is fully topological. Such a projection can be implemented by gauging the contractible higher-form symmetry supported on contractible submanifolds. We will call higher-form symmetry on contractible submanifolds the *contractible higher-form symmetry*. Since contractible higher-form symmetry acts trivially on the ground states without excitations, they are free of anomalies and can always be gauged. On the other hand, they are nontrivial when considered as symmetries that act on the full Hilbert space so gauging them is not a trivial operation.

Let us demonstrate the procedure using the toric code model on the square lattice. Each edge has a qubit, and the Hamiltonian terms are given by the product of four Pauli $X$ on edges that meet each vertex, and the product of four Pauli $Z$ on edges that surround each square (see the upper left-hand corner of Figure 2). The two types of excitations that violate these Hamiltonian terms are called electric charge and magnetic flux excitations, respectively.

---

[1]In our discussion we will focus on models with fully mobile excitations, and where the underlying one-form symmetry in field theory is topological with respect to deformations in any direction. Also, when we use the word "fully topological" for symmetry on the lattice, we mean the symmetry generator is invariant under small deformations on the lattice, which depends on the lattice geometry (for instance, on a square lattice, the smallest deformation of a loop is a small square).

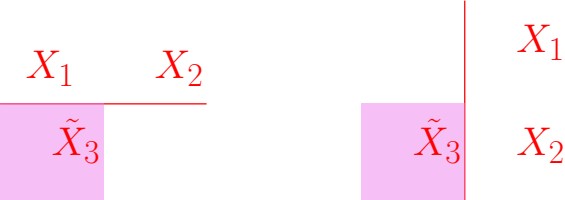

Figure 1: Gauss law constraints $X_1 X_2 \widetilde{X}_3 = 1$ for gauging the contractible one-form symmetry of the $m$-loop in the toric code model. By taking a product of the two terms to cancel $\widetilde{X}^2 = 1$, the Gauss law implies that the small contractible $m$-loop $\prod X$ supported on the four edges surrounding a vertex equals 1.

The theory has one-form symmetry generated by the magnetic loop operators, which run along the edges of the dual lattice and are given by the product of $X$ on the edges that intersect the dual loop. The one-form symmetry is not fully topological: a small loop can act nontrivially if the vertices surrounded by the loop support charge excitations. To obtain a model where the one-form symmetry is topological, we can project out the electric charge as follows. We introduce new $\mathbb{Z}_2$ gauge fields on the faces, with Pauli operators $\widetilde{X}, \widetilde{Y}, \widetilde{Z}$, and impose the Gauss law constraints shown in Figure 1. For the Hamiltonian to commute with the Gauss law constraints, we modify each flux term $\prod Z$ with four additional $\widetilde{Z}$ (see the upper right of Figure 2). On an infinite planar lattice, using the Gauss law constraints we can project out and remove the edge qubits, to arrive at a new model with face qubits only.[2] The small loop of $X$ in the original model becomes trivial in the new model since when two faces overlap the associated operators cancel $\widetilde{X}^2 = 1$, while a pair of adjacent large $X$ loops becomes a single large $\widetilde{X}$ loop, and the one-form symmetry is fully topological (see the lower part of Figure 2). We note that on an infinite plane, the Hamiltonian terms of the new model are the product of four $\widetilde{Z}$ on the faces surrounding each vertex [44].

The Gauss law in Figure 2 is different from the usual Gauss law imposed when gauging ordinary one-form symmetry (see e.g. Refs. [45, 46]). In the ordinary case, the Gauss law term is $\widehat{X}_p X_e \widehat{X}_{p'}$ for every edge $e$, and the two adjacent plaquettes $p, p'$. Such a Gauss law constraint is stronger than the one imposed in Figure 2. The latter constraint does not imply that all closed $X$-loops become trivial, while the former does. The $X$-loops that become trivial under the former Gauss law constraints are precisely those that are contractible, i.e. formed by product of the Hamiltonian star terms.

Since the original one-form symmetry on non-contractible cycles is still nontrivial after "gauging the contractible one-form symmetry", the resulting theory still has anomalous $\mathbb{Z}_2 \times \mathbb{Z}_2$ one-form symmetry, which guarantees nontrivial ground state degeneracy.

**Symmetries in Field Theories and Lattice Models.** In this work, we relate field theory and lattice models. In our examples, we start from field theories for topological quantum field theories (TQFTs), and we construct lattice models whose ground states describe the TQFT using the methods of e.g. Refs. [47–49]. Such lattice models typically have higher-form symmetries that are not fully topological, but only topological on the ground states. To obtain a lattice model with fully topological higher-form symmetry, we gauge the contractible higher-form symmetry as above. For instance, the $\mathbb{Z}_2$ gauge theory describes the ground states of the $\mathbb{Z}_2$ toric code. While the field theory has fully topological $\mathbb{Z}_2$ one-form symmetry generated by the magnetic flux, the toric code model does not have fully topological symmetry: the fully topological symmetry arises after gauging the contractible part of the symmetry.

---

[2]On a more general lattice, we cannot gauge-fix the edge qubit completely.

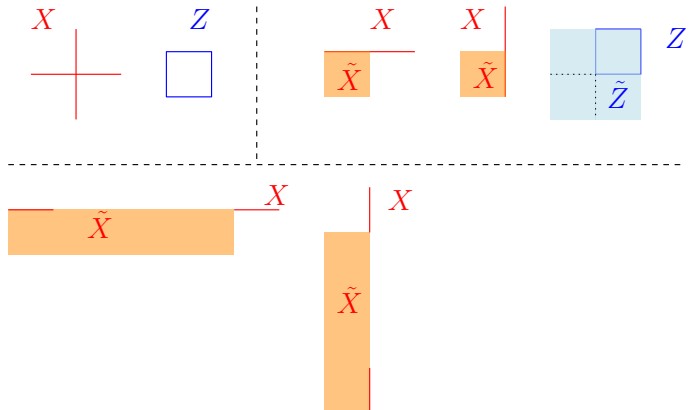

Figure 2: The upper figures are the Hamiltonian terms before (left) and after (right) gauging the contractible one-form symmetry generated by $\prod X$ over small loops on the dual lattice, with Gauss law imposed exactly as in Figure 1. The resulting fully topological truncated symmetry after gauging the contractible symmetry is shown in the lower figure. We note that on a infinite plane, the fully topological symmetry obeys a "global relation": the product of the symmetry on all columns equals the product of the symmetry on all rows.

### 1.1.2 Foliation dependence as new symmetry fractionalization

Starting with a fully topological higher $n$-form symmetry, we can construct a new $(n-1)$-form symmetry. The new symmetry is supported on a submanifold $\Sigma$ of codimension $n$, and here we consider $\Sigma$ that admit a foliation with codimension-$(n+1)$ leaves. If the spacetime dimension $D$ is sufficiently small $D \leq n+3$, $\Sigma$ always admits such a foliation. The new $(n-1)$-form symmetry generator is defined by the $n$-form symmetry on the leaf of the foliation.[3] When $\Sigma$ is contractible, the new $(n-1)$-form symmetry is trivial since the $n$-form symmetry is fully topological. Therefore the new $(n-1)$-form symmetry is also fully topological (invariant under continuous deformation).

The definition of the new $(n-1)$-form symmetry depends on the choice of foliation of $\Sigma$. For $\Sigma$ with only trivial codimension-1 cycles (with respect to $\Sigma$), the leaves of different foliations can be continuously deformed into each other. Furthermore, since the fully topological $n$-form symmetry on any contractible submanifolds $\delta$ is trivial, the new $(n-1)$-form symmetry on such a submanifold $\Sigma$ does not depend on the foliation. Such independence can be regarded as arising due to *global relations among the n-form symmetry generators*. To compare different foliations, we need to include different subgroups of one-form symmetry for the different foliations. For instance, for $n=1$ one-form symmetry, we need to turn on

$$B_2 = \sum_k v^k B_2^k,\tag{4}$$

where $k$ labels different foliations (restricted to the codimension-one 0-form symmetry generator), and $v^k B_2^k$ (without summation over $k$) describes the subgroup one-form symmetry for the foliation $k$.

**Violation of Foliation Independence as Fractionalization of Global Relation.** Foliation-independence can be violated in the presence of other operator insertions. When an $n$-dimensional operator $W$ that carries charge under $n$-form symmetry pierces the support $\Sigma$

---

[3]This is not to be confused with foliated symmetry: the new symmetry is not foliated because $\Sigma$ does not need to be the leaf of a foliation of the entire spacetime manifold.

of the new $(n-1)$-form symmetry, the contractible submanifolds $\delta$ that braid with $W$ are no longer trivial, and thus different foliations related by deforming their leaves with $\delta$ differ by the braiding of the $n$-form symmetry with $W$. Such a violation of foliation independence can be regarded as a new symmetry fractionalization of the $n$-form symmetry that violates the global relation. For instance, in 2+1D and for $n=1$, the violation of foliation-independence reproduces the fractionalization of global relations discussed in Ref. [26].

## 1.2 Outline

The paper is laid out as follows. In Section 2 we comment on global aspects of subsystem symmetries acting on lattice systems. In Section 3 we discuss subsystem symmetry fractionalization in 2+1D $\mathbb{Z}_2$ lattice gauge theory. In Section 4 we present a prescription for constructing lattice models with subsystem symmetry fractionalization inherited from a 1-form symmetry. In Section 5 we discuss the fractionalization of subsystem symmetries on loop and particle excitations in 3+1D $\mathbb{Z}_2$ lattice gauge theory. In Section 6 we introduce modified string-net lattice models that exhibit linear subsystem symmetry fractionalization in 2+1D and 3+1D. In Section 7 we conclude and list open questions. In Appendix A we review foliated gauge fields and their transition functions.

# 2 Global structure of subsystem symmetry on square lattice

In this section, we discuss how subsystem symmetry arises from one-form symmetry, and the global relation between subsystem symmetry generators. We also construct models where the global relations between subsystem symmetry generators are violated by excitations. This is a fractionalization of the global relation.

## 2.1 Global relation for subsystem symmetry

We consider Euclidean spacetime in 2+1D on a square spatial lattice with coordinate $(x,y)$ and time coordinate $t$. For a subsystem symmetry transformation generated by an operator supported on a line, say along the $x$ direction at fixed $y,t$, we can express the symmetry charge as

$$Q^y = \int dx\, j_0^y(x,y,t).$$ (5)

For the charge to be conserved, $\partial_t Q^y = 0$, the current satisfies the conservation law $\partial_0 j_0^y + \partial_x j_x^y = 0$. Similarly, we define charge $Q^x$ and current $(j_0^x, j_y^x)$.

To enforce a global relation we consider charges that satisfy a global constraint

$$\int dx\, Q^x = \int dy\, Q^y,$$ (6)

which implies

$$\int dx\, dy\, (j_0^x - j_0^y) = 0.$$ (7)

This means that the symmetry has a quotient: we denote the symmetry group of each subsystem symmetry by $G$, then the total symmetry is

$$\frac{G^{L_x} \times G^{L_y}}{G},$$ (8)

where $L_x, L_y$ are the linear sizes of the systems in the $x, y$ directions and describe the number of linear symmetries; the quotient identifies their diagonal subgroup as in Eq. (6). The quotient is generated by the symmetry with current

$$J_0 := j_0^x - j_0^y, \qquad J_x := j_x^y, \qquad J_y := -j_y^x, \qquad \partial_0 J_0 + \partial_x J_x + \partial_y J_y = 0. \tag{9}$$

In other words, by gauging the symmetry we impose the global relation. We remark that since the relation demands that product of lines in $x$ directions to be equal to product of lines in $y$ directions, while individually a line is not contractable, the product of suitable $x, y$ lines are contractible, and they can be expressed as product of small contractible loops.

In our description of the subsystem symmetry above, every generator is supported on a line. Hence we can think of the subsystem symmetry as a rigid subgroup of the one-form symmetry in the $x, y$ directions, not taking into account the global relation among the symmetry generators. Thus, we can say the subsystem symmetry is given by the rigid subgroup of the one-form symmetry subject to a constraint due to the global relation.

**Applications.** We remark that the formalism discussed here applies to any quantum system with higher-form symmetry. For instance, we can enrich $SO(3)$ Yang-Mills theory or $\mathbb{CP}^n$ sigma model in 3+1D with fractionalized subsystem symmetry.

## 2.2 Background gauge field

$\left(G^{L_x} \times G^{L_y}\right)$ **bundle: foliated two-form gauge field.** The background gauge fields couple to the current as

$$\int dt\, dx\, dy \left(A_\mu^x j_\mu^x + A_\mu^y j_\mu^y\right), \tag{10}$$

where $A^x = (A_0^x, A_y^x)$ and $A^y = (A_0^y, A_x^y)$. The conservation equations $\partial_0 j_0^x + \partial_y j_y^x = 0$ imply that the gauge fields have the gauge transformation

$$A^x \to A^x + d\lambda^x, \qquad A^y \to A^y + d\lambda^y, \tag{11}$$

where $\lambda^x = \lambda^x(t, y)$ and $\lambda^y = \lambda^y(t, x)$ to the maintain the vanishing components $A_x^x = 0$ and $A_y^y = 0$ that do not couple to the current. We note that $j_0^x$ can have step function discontinuities in $x$, and $j_0^y$ can have step function discontinuities in $y$. Thus the gauge fields $A^x, A^y$ can have delta function singularities in $x, y$ respectively.

We can describe such gauge fields $A^x, A^y$ using foliated two-form gauge fields [38, 39]

$$B_2^k = A^k e^k, \tag{12}$$

where $e^1 = dx, e^2 = dy$. These foliated gauge fields satisfy the constraint $B_2^k e^k = 0$, and can have delta function singularity in $x^k$ ($x^1 = x, x^2 = y$) [39]. Such foliated two-form background gauge fields describe $G^{L_x} \times G^{L_y}$ bundles for the linear subsystem symmetries.

At the same time, the foliated two-form gauge field describes the rigid subgroup of the one-form symmetry for generators supported on lines in the $x, y$ directions. The two-form currents are $J^k = j^k e^k$: $J^x = (j_0^x dt\, dx, j_y^x dy\, dx)$ and $J^y = (j_0^y dt\, dy, j_x^y dx\, dy)$.

$\left(G^{L_x} \times G^{L_y}\right)/G$ **bundle.** At this point we have not yet imposed the global relation that leads to the quotient in Eq. (8). This requires the background foliated two-form gauge fields to be subject to additional gauge transformations:

$$A^x \to A^x + h(x, y) f(t) dt, \qquad A^y \to A^y - h(x, y) f(t) dt, \tag{13}$$

where $h(x, y)$ is a step function in $x, y$ (if we consider the relation on a finite region). Let us discuss the values of possible parameters:

- If the gauge bundle is $G^{L_x} \times G^{L_y}$, the gauge transformation preserves the holonomy of $A$, and we require $\oint f(t) dt \in 2\pi\mathbb{Z}$.

- More generally, the parameters that satisfy $\int f(t) dt \notin 2\pi\mathbb{Z}$ are not the background gauge transformations of $G^{L_x} \times G^{L_y}$ bundle. They transform the Wilson lines of $A$ by a nontrivial phase. Since such a parameter transforms the Wilson line, they represent additional one-form gauge transformation on the $G^{L_x} \times G^{L_y}$ background gauge field. Equivalently, the bundle with the above transition function is a $\left(G^{L_x} \times G^{L_y}\right)/G$ bundle.

## 2.3 Fractionalization in gauge theory

We now consider $G = U(1)$ as a basic example where the global relation is fractionalized on a particle. We take the particle to be the electric charge of a $U(1)$ gauge theory with dynamical gauge field $a$.

Consider the particle sitting at the origin $(x, y) = (0, 0)$, described by $\oint a_0 dt$ along the temporal direction. The action is modified as

$$\int dt dx dy \left(A_0^x j^x + A_0^y j^y + a_0 \delta(x)\delta(y)\right). \tag{14}$$

Then the fractionalization of the global relation with integer $q$,

$$\int dx dy (j_0^x - j_0^y) = q \neq 0, \tag{15}$$

means that the gauge transformation in Eq. (13) acts on the dynamical gauge field $a$ describing the particle as

$$A^x \to A^x + f(t) dt, \qquad A^y \to A^y - f(t) dt, \qquad a \to a - q f(t) dt. \tag{16}$$

We remark that the transformation on the dynamical gauge field $a$ is a one-form transformation: under the shift by $f(t) dt$, the Wilson line of $a$ transforms as

$$W(\gamma) = e^{i\oint_\gamma a}, \qquad W(\gamma) \to e^{-iq\oint_\gamma f(t) dt} W(\gamma). \tag{17}$$

Such a correlated gauge transformation between the background fields and the dynamical gauge field $a$ represents a $\left(H_{\text{gauge}} \times G^{L_x} \times G^{L_y}\right)/G$ bundle, where $H_{\text{gauge}} = U(1)$ is the gauge symmetry for the dynamical gauge field $a$, and the quotient acts on $H_{\text{gauge}}$ by the common center of $G$ and $H_{\text{gauge}}$.

We remark that this is a direct analog of the fractionalization of conventional $G$ symmetry on the electric charge of dynamical $H$ gauge theory, where the fractionalization is described by a group $K$ which is an extension of $G$ by $H$ [2, 4, 21, 50].

## 2.4 Higher-group obstruction to symmetry fractionalization

**Review of conventional two-group symmetry.** We consider a gauge theory with gauge group, $H$, and matter fields that transform under flavor symmetry $G = \widetilde{G}/C$ with $C \subset Z(\widetilde{G})$. If $C$ can be identified with a gauge rotation in the center $Z(H)$, then the theory has one-form symmetry $\mathcal{A} = Z(H)/C$ that acts on the Wilson lines that cannot be screened by the matter fields.

Moreover, if $Z(H)$ is a non-split extension of $C$, such as $Z(H) = \mathbb{Z}_4$ and $C = \mathbb{Z}_2$, then the one-form symmetry and ordinary symmetry $G$ form a two-group symmetry that mixes the two symmetries [2]. The mixing is described by Postnikov class of the two-group

$$\Theta = \text{Bock}(w_2^f), \tag{18}$$

where $w_2^f$ is the obstruction class to lifting the $G$ bundle to a $\widetilde{G}$ bundle, and Bock is the Bockstein homomorphism for the short exact sequence

$$1 \to \mathcal{A} = Z(H)/C \to Z(H) \to C \to 1 \,.$$

This implies that one cannot gauge the ordinary symmetry without also gauging the one-form symmetry, which is known as the $H^3$ obstruction to symmetry fractionalization [4].

**Two-group involving fractionalization of global relation.** A similar discussion holds for the fractionalization of the global relation. For instance, consider the gauge group $H = \mathbb{Z}_4$, with the global relation over the Wilson line with odd charge fractionalized by a sign. In this theory, there is a two-group symmetry that combines $\mathbb{Z}_2$ one-form symmetry and the subsystem symmetry. The analog of the Postnikov class is

$$\Theta = \text{Bock}(w_2^f) \,, \tag{19}$$

where $w_2^f$ is the obstruction to lifting the $\left(G^{L_x} \times G^{L_y}\right)/\mathbb{Z}_2$ bundle to $\left(G^{L_x} \times G^{L_y}\right)$ bundle. If $G$ is an Abelian group, the Postnikov class above is in fact trivial, and thus there is no $H^3$ obstruction to symmetry fractionalization. On the other hand, if $G$ is non-Abelian such as $G = D_8$ the dihedral group of order 8 or $G = SU(2)$), then the above example has non-trivial $H^3$ obstruction to symmetry fractionalization.

## 2.5 Anomaly of subsystem symmetry fractionalization

We now investigate the anomaly of the subsystem symmetry by coupling the system to a nontrivial background gauge field. If there is an inconsistency that requires a non-trivial bulk, then the symmetry is anomalous. See also previous related work in Ref. [51].

We focus on the background field configuration where there is no global relation. If there is an anomaly for such fields, then the full subsystem symmetry must be anomalous. Such field configurations can be described by the background of a one-form symmetry.

We denote the one-form symmetry by $\mathcal{A}$, the background field for the one-form symmetry by $B$, and homomorphism $v : G \to \mathcal{A}$. Then coupling to the background fields of a subsystem without global relation is equivalent to setting

$$B = v(B^x + B^y) \,, \tag{20}$$

where $B^x = A^x dx, B^y = A^y dy$ are the foliated two-form gauge fields describing the rigid subgroup one-form symmetry in the $x, y$ directions, respectively. Following the discussion in Section 2.3, Eq. (20) implies that after we constrain the backgrounds $B^x, B^y$ to satisfy the global relation, the global relation will be fractionalized on the particles that transform under the one-form symmetry.

The anomaly of the full one-form symmetry can be described by the statistics of the generator

$$\theta : \mathcal{A} \to U(1) \cong \mathbb{R}/2\pi\mathbb{Z} \,. \tag{21}$$

Then the anomaly of the one-form symmetry is described by the bulk SPT phase with effective action [52]

$$2\pi \int \theta[B] \,, \tag{22}$$

where the function is composed with the generalized Pontryagin square quadratic operation to obtain a four-form from two-form $B$.

The anomaly of the rigid subgroup one-form symmetry is given by substituting (20) into (22): using $B^x B^x = 0$ and $B^y B^y = 0$, we find the anomaly

$$2\pi \int Q[B^x \cup B^y] := 2\pi \int \left( \theta[B^x + B^y] - \theta[B^x] - \theta[B^y] \right). \tag{23}$$

This means that when the generators of the one-form symmetry have non-trivial mutual braiding, the subsystem symmetry must be anomalous.

For instance, when the one-form symmetry is $\mathbb{Z}_2$, and the subsystem symmetry is also $\mathbb{Z}_2$, then if the one-form symmetry is generated by a semion or antisemion, the subsystem symmetry is anomalous. The anomaly is described by the bulk term

$$\frac{2}{2\pi} \int B^x B^y, \tag{24}$$

which describes a non-trivial subsystem symmetry-protected topological (SSPT) phase in the bulk. Such a bulk theory is discussed in Ref. [39].

# 3 Example: $\mathbb{Z}_2$ gauge theory in 2+1D

In this section we consider $H = \mathbb{Z}_2$ gauge theory with a linear subsystem symmetry group $G = \mathbb{Z}_2$.

The $\mathbb{Z}_2$ gauge theory without subsystem symmetry is be described by

$$\frac{2}{2\pi} a \, d b, \tag{25}$$

where $a, b$ are one-form gauge fields, and the Wilson line is $e^{i \oint a}$, while the magnetic line is $e^{i \oint b}$. The magnetic line carries nontrivial holonomy of the $\mathbb{Z}_2$ gauge field, and the magnetic line braids with the Wilson line by the Aharonov-Bohm phase.

The theory has a one-form symmetry that acts on the $\mathbb{Z}_2$ Wilson line. Since the magnetic line braids with the Wilson line, the generator of such one-form symmetry is the magnetic line.[4]

In the following, we will discuss the subsystem symmetry given by the subgroup of the relativistic one-form symmetry generated by a subset of magnetic lines. For relativistic symmetries, the only global relation between the symmetry generators come from the global topology when the commutators of the generators can be contracted to a point. In our case, we will consider subsystem symmetries that obey additional global relations that are not present for the fully topological relativistic symmetries. The difference between subsystem and relativistic relations is those additional relations, which can be imposed as additional Gauss constraints.

## 3.1 Subsystem symmetry without global relation

We begin with the subsystem symmetry without the global relation. Then the subsystem symmetry is simply a rigid subgroup of the one-form symmetry. The background of such a subgroup corresponds to the configuration

$$B = B^x + B^y, \tag{26}$$

---

[4]The analog of such a magnetic line operator in 3+1D is a surface operator. This is similar to the 't Hooft lines of continuous gauge fields, where the monopole carries flux on the surrounding sphere. Here, the the magnetic lines carry holonomy on the surrounding circle.

where $B$ is the background for the full one-form symmetry, and $B^x = A^x dx$, $B^y = A^y dy$ are the background for the linear symmetries in the $x, y$ directions. We remark that the background gauge fields that are exact, i.e. equivalent to a one-form gauge transformation, describe a symmetry generated by contractible loops.

### 3.1.1  Gauging $\mathbb{Z}_2$ subsystem symmetry

The $\mathbb{Z}_2$ gauge theory coupled to background $B$ can be described by

$$\frac{2}{2\pi} a db + \frac{2}{2\pi} bB, \tag{27}$$

where $b$ is a Lagrangian multiplier that enforces $da + B = 0$.

For $B = B^x + B^y$, gauging the subsystem symmetry gives

$$\frac{2}{2\pi} a db + \frac{2}{2\pi} \sum_{k=x,y} bB^k + \sum_{k=x,y} \frac{2}{2\pi} B^k dA^k, \tag{28}$$

where we include a Lagrangian multiplier scalar $A^k$ to enforce $B^k$ takes $\mathbb{Z}_2$ value.

The gauge transformation is

$$B^k \to B^k + d\lambda^k, \qquad\qquad A^k \to A^k + f(x^k) - \lambda,$$
$$a \to a + d\beta - \sum_{k=x,y} \lambda^k, \qquad b \to b + d\lambda, \tag{29}$$

where $\lambda^k$ is a one-form that only has $dx^k$ component, and $\lambda, \beta, f$ are scalars. In particular, the scalar $\varphi \equiv A^x - A^y$ has the gauge transformation $\varphi \to \varphi + f(x) + g(y)$, which can be viewed as a $\mathbb{Z}_2$ version of the scalar describing the XY plaquette model. The Wilson line $e^{i \oint_\gamma a}$ is gauge invariant only when it lies entirely along the time direction, and thus it describes a fracton.

We note that the theory $A^k, B^k$ describes layers of $\mathbb{Z}_2$ symmetry breaking state in 1+1D theory of two vacua. Each layer has a $\mathbb{Z}_2$ symmetry that exchanges the two vacua, and coupling to $b$ gauges the diagonal $\mathbb{Z}_2$ symmetry of the $x$ and $y$ layers.

## 3.2  2+1D Lattice models

In this section, we construct lattice models in 2+1D where the global relation between the symmetry lines in the $x, y$ directions are only realized on ground states. In this case, the subsystem symmetry is extended. Later on, we gauge a symmetry such that the topological order is not changed, but the subsystem symmetry along $x, y$ directions becomes fractionalized. After gauging the symmetry, the new model coincides with the model in Ref. [26] with fractionalized global relations for subsystem symmetry.

### 3.2.1  Model with $G^{L_x} \times G^{L_y}$ symmetry

**$\mathbb{Z}_2$ global $\times$ subsystem symmetry protected topological phase.**   We now construct a lattice model with subsystem fractionalization. We start with an SSPT phase with global $\mathbb{Z}_2$ symmetry and $\mathbb{Z}_2^x \times \mathbb{Z}_2^y$ subsystem symmetry and gauge the $\mathbb{Z}_2$ global symmetry. The SSPT phase is described by the cocycle (where we normalize the variables to take discrete value 0, 1 mod 2)

$$\phi_3(b, B^k) = \sum_k b \cup B^k. \tag{30}$$

It satisfies

$$\phi_3(d\lambda, d\lambda^k) = d\phi_2, \qquad \phi_2 = \sum_k \lambda \cup d\lambda^k. \tag{31}$$

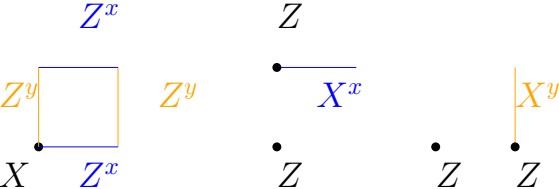

Figure 3: Local Hamiltonian terms for a SSPT phase with $\mathbb{Z}_2$ ordinary and $\mathbb{Z}_2$ subsystem symmetry introduced in Eq. (32).

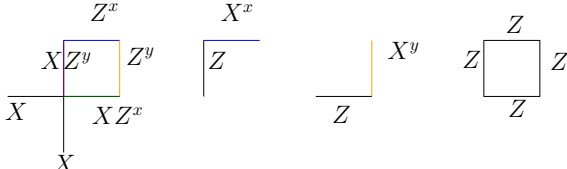

Figure 4: Local terms appearing in the Hamiltonian for $\mathbb{Z}_2$ ordinary gauge theory with $\mathbb{Z}_2$ subsystem symmetries along $x, y$ directions, see Eq. (33).

We introduce one qubit at each vertex, which is acted on by Pauli matrices $X, Y, Z$. We also introduce one qubit on each edge: on the edge in the $x$ direction, we introduce an $x$ qubit, and on the edge in the $y$ direction we introduce a $y$ qubit. They are acted on by Pauli matrices $X^x, Y^x, Z^x$ and $X^y, Y^y, Z^y$. Denote $\lambda = (1 - Z)/2$, $\lambda^k = (1 - Z^k)/2$ where $k = x, y$. The Hamiltonian for the SSPT phase is obtained by conjugating the trivial Hamiltonian $H_0 = -\sum X - \sum_{k=x,y} \sum X^k$ by the diagonal unitary $(-1)^{\int \phi_2}$:

$$H_{\text{SSPT}} = -\sum_v X_v (-1)^{\int \widetilde{v} \cup (d\lambda^x + d\lambda^y)} - \sum_{e_x} X^x_{e_x} (-1)^{\int d\lambda \cup \widetilde{e}_x} - \sum_{e_y} X^y_{e_y} (-1)^{\int d\lambda \cup \widetilde{e}_y}, \tag{32}$$

where $e_x, e_y$ are edges in the $x, y$ directions, and $\widetilde{e}$ is the one-cochain that takes value 1 on edge $e$ and zero on all other edges. The terms in the Hamiltonian are shown in Figure 3.

**$\mathbb{Z}_2$ Toric code enriched by subsystem symmetry.** We now gauge the conventional global symmetry in the model introduced above. To achieve this we introduce one gauge qubit on each edge, with $b = (1 - Z_e)/2$, gauge fix the degrees of freedom at each vertex $v$, and add a flux term for the $b$ gauge qubits:

$$H = -\sum_v \left( \prod X_e \right) (-1)^{\int \widetilde{v} \cup (d\lambda^x + d\lambda^y)} - \sum_{e_x} X^x_{e_x} (-1)^{\int b \cup \widetilde{e}_x} - \sum_{e_y} X^y_{e_y} (-1)^{\int b \cup \widetilde{e}_y} - \sum_f \left( \prod Z_e \right). \tag{33}$$

The terms in the Hamiltonian are shown in Figure 4. This theory has linear subsystem symmetries generated by $\prod X^y$ along the $x$ direction on the dual lattice, and $\prod X^x$ along the $y$ direction on the dual lattice.

We remark that the group generated by neighboring pairs of linear symmetries along the same direction has a global relation: the product over all pairs gives the identity. This global relation has non-trivial fractionalization: the product of a finite number of pairs on a ribbon-like region of finite width is equivalent to the product of two Wilson lines $\prod Z$ on either side of the ribbon due to the $ZX^x$ and $ZX^y$ stabilizers. Thus the global relation for the group generated by pairwise linear symmetries in the same directions has fractionalization given by a sign on the $m, f$ particles that braid nontrivially with the Wilson line $\prod Z$.

**Gauging both global and subsystem symmetry in SSPT.** To simultaneously gauge the global and subsystem symmetry of the Hamiltonian in Eq.(32) we introduce two qubits on

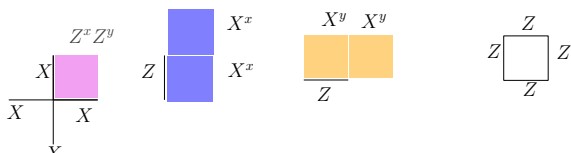

Figure 5: Local terms in the Hamiltonian for the fully gauged SSPT phase introduced in Eq. (34).

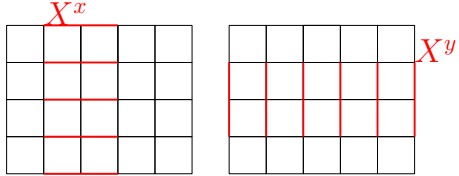

Figure 6: Linear symmetries in the $x, y$ directions given by pairs of $X^x$-lines and $X^y$-lines.

each face, which are acted on by $X_f^k, Y_f^k, Z_f^k$ with $k = x, y$. Let $B_f^k = (1 - Z_f^k)/2$, then the gauged Hamiltonian is:

$$
\begin{aligned}
H' = &-\sum_v \left( \prod X_e \right) (-1)^{\int \widetilde{v} \cup (B^x + B^y)} - \sum_{e_x} \left( \prod X_f^x \right) (-1)^{\int b \cup \widetilde{e}_x} \\
&- \sum_{e_y} \left( \prod X_f^y \right) (-1)^{\int b \cup \widetilde{e}_y} - \sum_f \left( \prod Z_e \right),
\end{aligned}
\tag{34}
$$

the terms in this Hamiltonian are shown in Figure 5. The ground state degeneracy of this Hamiltonian on the torus of size $L_x, L_y$ is:

$$
\text{GSD} = 2^{L_x + L_y - 1},
\tag{35}
$$

which was obtained by following the method of Refs. [53, 54] The model with gauged subsystem symmetry has excitations with restricted mobility [49].

### 3.2.2 Model with $\left( G^{L_x} \times G^{L_y} \right)/G$ symmetry: Fractionalized global relations for $x, y$ linear symmetries

In the model without fractionalized global relation between the $x, y$ linear symmetry, there are symmetries generated by a pair of $\prod X^x$ along a line in the $y$ direction on the dual lattice, and similar symmetry along the $x$ direction (see Figure 6).

To obtain a model with $\left( G^{L_x} \times G^{L_y} \right)/G$ linear subsystem symmetry with a global relation, we can gauge the diagonal subgroup given by the product of all the above linear symmetries in the $x, y$ directions. We achieve this via the following steps:

- We introduce a new qubit on each face, and impose the Gauss law terms in Figure 7.

Figure 7: Gauss law terms that appear after gauging the global diagonal subgroup of $\left( G^{L_x} \times G^{L_y} \right)$ which produces a subsystem symmetry with a global relation.

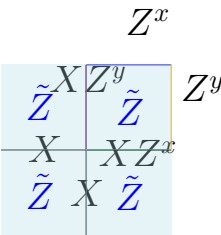

Figure 8: Modified vertex term in Figure 4 with "Wilson surfaces" $\widetilde{Z}$ attached to ensure commutativity with the Gauss law constraint depicted in Figure 7.

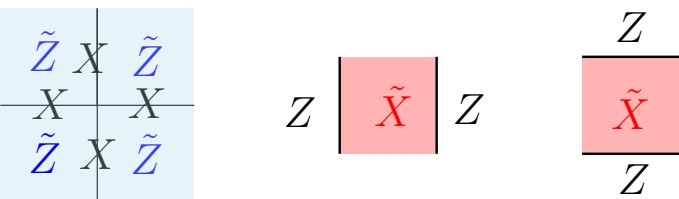

Figure 9: Local Hamiltonian terms from a model with $\left(G^{L_x} \times G^{L_y}\right)/G$ subsystem symmetry for $G = \mathbb{Z}_2$ and a fractionalized global relation. This model is obtained as follows: (1) we start with toric code enriched with subsystem one-form symmetry (obtained by gauging the $\mathbb{Z}_2$ symmetry in an SSPT phase), and (2) gauge the contractible part of the subsystem one-form symmetry to obtain subsystem one-form symmetry with global relations.

- We modify the vertex term in the Hamiltonian of Figure 4 with a "Wilson surface" $\widetilde{Z}$ shown in Figure 8 to ensure that the Hamiltonian commutes with the Gauss law term shown in Figure 7.

- We gauge-fix the edge qubits $Z^x = 1, Z^y = 1$. This produces a Hamiltonian with $\left(G^{L_x} \times G^{L_y}\right)/G$ subsystem symmetry for $G = \mathbb{Z}_2$. The local terms of this Hamiltonian are depicted in Figure 9. We remark the plaquette $\prod Z$ flux terms do not appear directly as they can be obtained from the product of the second and third terms in Figure 9. The Hamiltonian model reproduces the lattice model on the dual lattice with fractionalized global relation for subsystem symmetry that was discussed previously in Ref. [26].

We remark that this model can be equivalently obtained by performing these steps in the reverse order. That is, we can start with an SSPT with a global $\mathbb{Z}_2$ symmetry and a $\mathbb{Z}_2^x \times \mathbb{Z}_2^y$ subsystem symmetry which already satisfies the global relation, and then simply gauge the global symmetry, as shown in Ref. [26].

## 3.3 Example with anomalous subsystem symmetry fractionalization

We now construct a lattice model on the 2+1D boundary of a 3+1D bulk, such that the boundary exhibits anomalous fractionalization of subsystem symmetry. Our approach is to first construct a model with subsystem symmetry $G^{L_x} \times G^{L_y}$ for $G = \mathbb{Z}_2$, and then obtain a model that has subsystem symmetry $\left(G^{L_x} \times G^{L_y}\right)/G$ with a global relation by gauging a suitable subgroup.

### 3.3.1 Model with $G^{L_x} \times G^{L_y}$ symmetry

We consider the subsystem symmetry protected topological phase in 3+1D defined by the cocycle

$$\phi_4 = \pi B^x B^y \,, \tag{36}$$

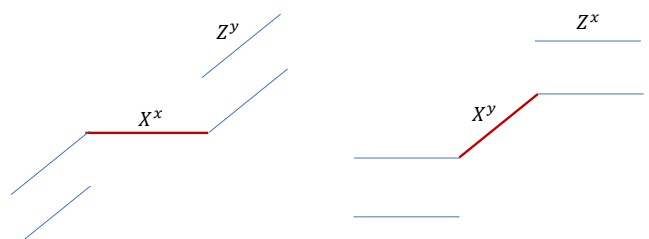

Figure 10: The Hamiltonian for the subsystem symmetry SPT phase in 3+1D from the one-form symmetry SPT.

where $B^x, B^y$ are the background for the subsystem symmetries on the $yz$-planes and $xz$-planes, respectively.

Following the method in Ref. [55], we can construct the Hamiltonian for a bulk SSPT phase as in Figure 10 (see also Ref. [39]). On each edge we introduce a qubit, which is acted on by $X, Y, Z$, and define $a = (1 - Z)/2$. The bulk wavefunction is

$$\Psi = \sum_{\{a\}} (-1)^{\int a^x \cup da_y} |\{a^x(e_x), a^y(e_y)\}\rangle, \tag{37}$$

where $e_x, e_y$ are edges in the $x, y$ directions. Here the field configuration $a$ is fixed on the boundary and summed over in the bulk. The 3+1D Hamiltonian is given by

$$H_{3+1D} = -\prod_{e_x} X^x_{e_x} (-1)^{\int \tilde{e}_x \cup da_y} - \prod_{e_y} X^y_{e_y} (-1)^{\int da_x \cup \tilde{e}_y}, \tag{38}$$

where $a_x = (1 - Z^x)/2$ and $a_y = (1 - Z^y)/2$, and $\tilde{e}$ is the one-cochain that takes value 1 on edge $e$ and 0 otherwise.

Under a one-form transformation $a^x \to a^x + \lambda^x$, $a^y \to a^y + \lambda^y$, for some cocycles $\lambda^x, \lambda^y$, the bulk wavefunction transforms by the boundary term

$$(-1)^{\int \lambda^x \cup a_y}. \tag{39}$$

A gapped boundary Hamiltonian is then given by

$$H = -\sum_v (-1)^{\int \tilde{v} \cup da_y} \prod X^x_{e_x} - \sum_v \prod X^y_{e_y} - \sum_f \prod Z^x_{e_x} - \sum \prod Z^y_{e_y}. \tag{40}$$

We note that the contractible part of the subsystem symmetries are nontrivial. For instance, the product of the subsystem symmetries in the $xz$ planes and $yz$ planes is

$$\prod X^x_{e_x} \prod X^y_{e_y}, \tag{41}$$

which is a non-trivial on the full Hilbert space, but acts trivially on the ground states.

### 3.3.2 Model with $\left(G^{L_x} \times G^{L_y}\right)/G$ symmetry

Let us modify the model by gauging the diagonal $G$ subgroup to obtain a theory with $\left(G^{L_x} \times G^{L_y}\right)/G$ symmetry, where the Gauss law constraints are as in Figure 7 with new variables on the plaquettes that lie in the xy-plane, acted on by Pauli operators $\tilde{X}, \tilde{Y}, \tilde{Z}$. The new model is shown in Figure 11.

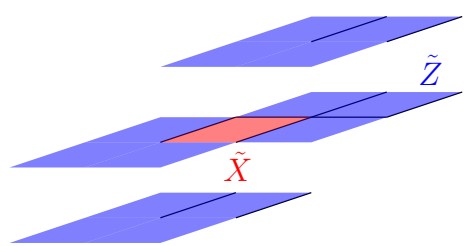

Figure 11: 3+1D Hamiltonian (upper) for an SSPT with subsystem one-from symmetry, after gauging the contractible one-form symmetry.

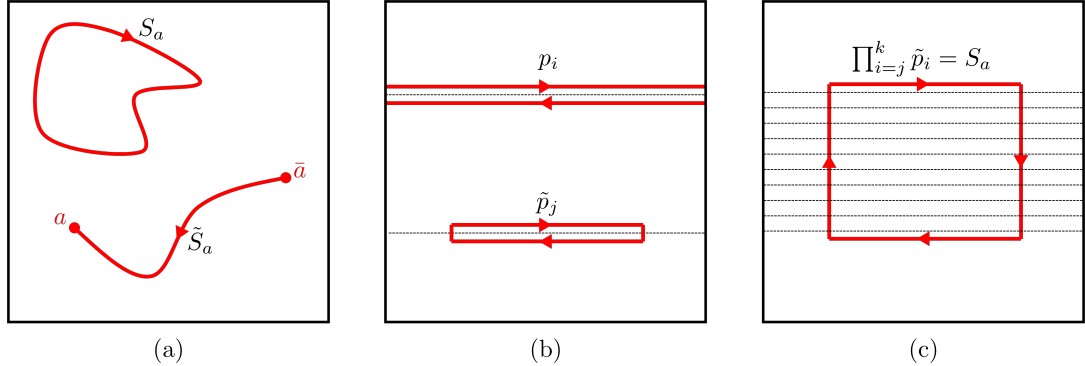

Figure 12: (a) Wilson loops in a topologically ordered phase act as one-form symmetries that create anyons at their endpoints. (b) The subsystem symmetry generator $p_i$ is a pair of oppositely oriented Wilson lines acting near row $i$ (this row is indicated by the dotted line). These pairs create trivial superselection sectors at their endpoints, so they can be truncated to a finite segment, giving the operators $\widetilde{p}_j$ which are essentially small Wilson loops. (c) Taking a product of $\widetilde{p}_j$ in a finite region equals a Wilson loop on the boundary of that region.

## 4 Generic construction of anomalous and non-anomalous lattice models

We describe a simple prescription for constructing lattice models for subsystem symmetry fractionalization using the logic introduced in the previous sections. That is, we consider lattice models for topological orders having one-form symmetries and then explicitly obtain the subsystem symmetry as a rigid subgroup of the one-form symmetry.[5]

The generic construction is depicted in Fig. 12. We consider a topological model with spontaneously broken one-form symmetry generated by Wilson lines whose endpoints create anyons. We then define a linear subsystem symmetry group formed by pairs of straight Wilson lines moving in the horizontal direction (with opposite orientations). Denote the subsystem symmetry generator corresponding to the pair of lines acting near row $i$ as $p_i$. These subsystem symmetries are unbroken in the topological phase: truncating them to a finite line segment creates topologically trivial excitations at the endpoints of the segment which can be removed by acting with local operators at the endpoints, see Fig. 12. This defines the localized symmetry generator $\widetilde{p}_i$.

These symmetries satisfy the global relation that says the product of all pairs is the identity, $\prod_i p_i = I$. If we restrict this global relation to a finite patch, we find that the product of all truncated symmetries in a patch, $\prod_{i=j}^k \widetilde{p}_i$, equals a Wilson loop circling the boundary of the

---

[5]We thank Michael Hermele for suggesting this construction.

patch, see Fig. 12. In Ref. [26], it was argued that the anyon type of this Wilson loop labels the fractionalization class of the global relation. By choosing different types of Wilson lines to form our subsystem symmetry group, we can therefore access different fractionalization classes corresponding to the same topological order.

One can also confirm that this construction reproduces the physical properties that are expected of models with subsystem symmetry fractionalization [26], such as the restricted mobility of anyons that braid non-trivially with the anyon labeling the fractionalization class.

## 4.1 Example: $\mathbb{Z}_2$ gauge theory

As an explicit example of the above construction, we apply it to $\mathbb{Z}_2$ gauge theory as captured by the toric code model. Here, there are three types of non-trivial Wilson lines we can use to construct the subsystem symmetry group: the electric and magnetic bosons and the fermion. In each case, we get a subsystem symmetry group of $\mathbb{Z}_2$ operators supported on lines, where the fractionalization class is labeled by the anyon associated to the chosen Wilson line. Importantly, each type of Wilson loop can be realized by a product of onsite operators in the toric code model, so these symmetries are not anomalous on the lattice.

## 4.2 Example: Anomalous model with semion fractionalization

We can also construct a model with $\mathbb{Z}_2$ linear subsystem symmetries where the fractionalization class is labeled by a semion. According to the discussion of the previous sections, this fractionalization class is anomalous. This anomaly is manifested in the fact that the linear subsystem symmetries cannot be made onsite.

Consider a topological order containing a $\mathbb{Z}_2$ semion, such as the double-semion model [56], and apply the construction outlined in this section to the Wilson lines corresponding to the semion to get a fractionalization class labeled by the semion. An important difference to the toric code case is that the Wilson lines for a semion cannot be written as a product of on-site operators due to the $H^3$ anomaly [57,58]. Therefore, the subsystem symmetries constructed from rigid semion Wilson lines are non-onsite, reflecting the fact that this fractionalization class is anomalous.

We remark that using the lattice model of Ref. [59], we can write semion Wilson lines which are onsite, but they become $\mathbb{Z}_4$ operators rather than $\mathbb{Z}_2$. Therefore, the fractionalization class labelled by a semion is not anomalous for $\mathbb{Z}_4$ subsystem symmetry.

What is the bulk 3+1D SSPT that matches this anomalous symmetry fractionalization? Observe that a semion string operator has the same structure as the boundary symmetry of the non-trivial 2+1D SPT phase with global $\mathbb{Z}_2$ symmetry, as can be explicitly observed using the Pauli stabilizer model derived in Ref. [59]. Take a stack of such 2+1D models in the z-direction, and consider the subsystem symmetry group generated by applying the global symmetry to a pair of neighboring layers in the stack. Under this symmetry, the stack of 2+1D SPTs is a non-trivial 3+1D SSPT phase. The action of this symmetry on the boundary of the stack (parallel to the z-direction) is equivalent to the action of pairs of rigid semion string operators, as in our anomalous 2+1D model. Therefore, we argue that our model of anomalous symmetry fractionalization can be put on the boundary of a 3+1D SSPT constructed from a stack of 2+1D $\mathbb{Z}_2$ SPTs.

# 5 Example: $\mathbb{Z}_2$ gauge theory in 3+1D

In this section, we consider $\mathbb{Z}_2$ gauge theory in 3+1D. The theory can be described by

$$\frac{2}{2\pi}adb\,,\tag{42}$$

where $a$ is a one-form, and $b$ is a two-form gauge field. The theory has $\mathbb{Z}_2$ Wilson line operator $e^{i\oint a}$ and magnetic surface operator $e^{i\oint b}$. They have mutual $(-1)$ braiding, and they generate two-form and one-form symmetries, respectively.

In the following, we discuss enriching the theory by subsystem symmetries from rigid subgroups of the one-form symmetry or two-form symmetry. We consider planar subsystem symmetries on planes with constant $x$, $y$, or $z$ space coordinate and linear subsystem symmetries on lines with constant $(x,y),(y,z)$ or $(x,z)$ space coordinate.

## 5.1 Global relations for subsystem symmetries

### 5.1.1 Linear subsystem symmetry and 2-foliated gauge field

The linear subsystem symmetry satisfies the same global relation as in 2+1D for each plane. For instance, the product of linear symmetries aligned with the $x$ direction, over a plane with varying $y$ coordinate, coincides with the product of the linear symmetries aligned with the $y$ direction, over a plane with varying $x$ coordinate,

$$\int Q^{yz}dy = \int Q^{xz}dx\,,\qquad \int j_0^{yz}dxdy = \int j_0^{xz}dydx\,.\tag{43}$$

**Background gauge field for $G^{L_x} \times G^{L_y}$.** The currents $j^{yz} = (j_0^{yz}, j_x^{yz})$ couple to the background gauge field $A^{yz} = (A_0^{yz}, A_x^{yz})$ as $\int dt dx dy dz \left(j_0^{yz}A_0^{yz} + j_x^{yz}A_x^{yz}\right)$. The background fields can be described by a 2-foliated background three-form gauge field

$$B_3^{kl} = A^{kl}dx^k dx^l\,.\tag{44}$$

The 2-foliated gauge field satisfies $B_3^{kl}e^k = 0 = B_3^{kl}e^l$, for $e^k = dx^k$. Such a background gauge field describes a rigid subgroup of the two-form symmetry.

**Background gauge field for $(G^{L_x} \times G^{L_y})/G$.** The global relation for linear subsystem symmetry $\int j_0^{yz}dxdy = \int j_0^{xz}dydx$ implies that the background fields have additional gauge transformation

$$A^{yz} \to A^{yz} + f(t,z)dt\,,\qquad A^{zx} \to A^{zx} - f(t,z)dt\,.\tag{45}$$

The background fields with such a transformation describe $(G^{L_x} \times G^{L_y})/G$ bundles.

### 5.1.2 Planar subsystem symmetry and 1-foliated gauge fields

The planar symmetry in 3+1D is similar to the linear symmetry in 2+1D, as both are subgroups of one-form symmetry. The charge of the planar symmetry on planes normal to the $x$ direction is given by $Q^x(x,t) = \int j_0^x dy dz$, and similarly for $Q^y, Q^z$.

Let us consider subsystem symmetry of three colors $x, y, z$, such that the generators of planar subsystem symmetry on $x, y$-planes have $x, y$ charges, and similar for other planar generators. Then the planar subsystem symmetry satisfies the following global relation:

$$\int Q^x dx + \int Q^y dy + \int Q^z dz = 0\,.\tag{46}$$

Such subsystem symmetry has a group structure $\left(G^{L_x} \times G^{L_y} \times G^{L_z}\right)/G$.

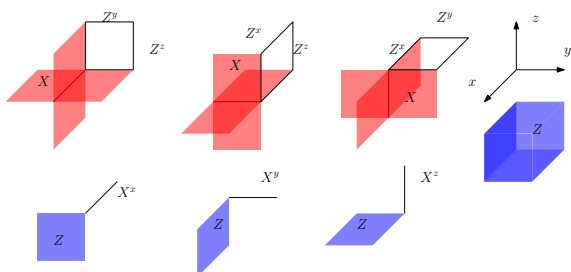

Figure 13: Local Hamiltonian terms of a 3+1D $\mathbb{Z}_2$ two-form gauge theory enriched by planar subsystem symmetry $G^{L_x} \times G^{L_y} \times G^{L_z}$ with no global relations between symmetries aligned with different directions.

**Background gauge field for $G^{L_x} \times G^{L_y} \times G^{L_z}$.** The background gauge fields couples as $\int dt\,dx\,dy\,dz \left( A_0^x j_0^x + A_y^x j_y^x + A_z^x j_z^x \right)$, and similarly for $j^y, j^z$. We can describe the background gauge fields using a 1-foliated two-form gauge field

$$B_2^k = A^k dx^k, \tag{47}$$

which satisfies $B_2^k e^k = 0$ for $e^k = dx^k$. Such background gauge fields describe a rigid subgroup of one-form symmetry.

**Background gauge field for $\left( G^{L_x} \times G^{L_y} \times G^{L_z} \right)/G$.** From the global relation

$$\int j_0^x dy\,dz\,dx + \int j_0^y dx\,dz\,dy + \int j_0^z dx\,dy\,dz = 0,$$

we are led to consider background gauge fields with additional gauge transformations

$$A^x \to A^x + f^x(t)dt, \qquad A^y \to A^y \to A^y + f^y(t)dt, \qquad A^z \to A^z - (f^x(t) + f^y(t))\,dt. \tag{48}$$

Such background gauge fields describe $(G^{L_x} \times G^{L_y} \times G^{L_z})/G$ bundles.

## 5.2 Fractionalization on particles

The planar subsystem symmetry can act nontrivially on the (fully mobile) electric particles as it is a subgroup of the magnetic one-form symmetry. On the other hand, the linear subsystem symmetry must act trivially on the (fully mobile) electric particles as it is a subgroup of the electric two-form symmetry. Thus it is only of interest to consider the fractionalization of planar subsystem symmetry.

### 5.2.1 Enriching with $(G^{L_x} \times G^{L_y} \times G^{L_z})$

Here, we enrich the toric code Hamiltonian in 3+1D that describes $\mathbb{Z}_2$ gauge theory with planar subsystem symmetry without global relations. In field theory, this means we turn on background gauge fields for the rigid subgroup one-form symmetry

$$B_2 = B_2^x + B_2^y + B_2^z. \tag{49}$$

The one-form symmetry in $\mathbb{Z}_2$ gauge theory is not anomalous, and thus the planar subsystem symmetry subgroup is also non-anomalous. We remark that if we gauge the subsystem symmetry, this produces the X cube model [37, 44, 60].

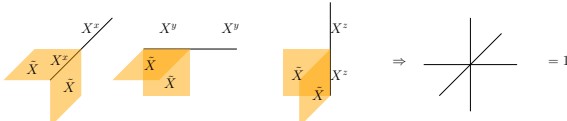

Figure 14: Gauss law constraints that arise when gauging the diagonal subgroup of the planar subsystem symmetry of the Hamiltonian in Eq.(51).

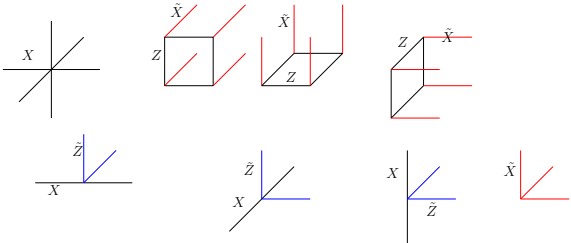

Figure 15: Local terms in the Hamiltonian with planar subsystem symmetry and a fractionalized global relation obtained by gauging the Hamiltonian in Eq.(51).

To obtain a lattice Hamiltonian model we use the toric code. We start with the SSPT phase with $\mathbb{Z}_2$ one-form symmetry and subsystem symmetry, and the cocycle

$$\phi_4 = b \cup \sum_k B_2^k, \tag{50}$$

and then gauge the one-form symmetry for $b$ to obtain $\mathbb{Z}_2$ two-form gauge theory, which is equivalent to $\mathbb{Z}_2$ one-form gauge theory in 3+1D.

Consider the cubic lattice with a local Hilbert space given by a qubit on each face for the $\mathbb{Z}_2$ two-form gauge field, and a qubit on edge for the transformation parameter of the subsystem symmetry. These degrees of freedom are acted on by the Pauli operators $X, Y, Z$ on faces, and $X_e^k, Y_e^k, Z_e^k$ on edges, respectively. This gives the stabilizer Hamiltonian model for $\mathbb{Z}_2$ gauge theory enriched by subsystem symmetry:

$$H = -\sum_e \left( \prod_{e \in \partial f} X_f \right) (-1)^{\int \widetilde{e} \cup \sum_k d\lambda_1^k} - \sum_c \prod_{f \in \partial c} Z_f - \sum_k \sum X_{e_k}^k (-1)^{\int b \cup \widetilde{e}_k}, \tag{51}$$

where the integral is over 3D space, $c$ are cubes, $e_k$ are edges in the $x^k$ direction, and $\widetilde{e}, \widetilde{e}_k$ are one-cochain that takes value 1 on edges $e, e_k$ and 0 on other edges. The Hamiltonian terms are depicted in Figure 13.

### 5.2.2 Enriching with $(G^{L_x} \times G^{L_y} \times G^{L_z})/G$: Fractionalized global relation

To construct a model enriched by planar subsystem symmetry that satisfies the desired global relations, we gauge the diagonal symmetry in the model above. The resulting gauged model inherits subsystem symmetry with fractionalized global relations.

To perform the gauging transformation we introduce a new gauge qubit on each face, acted on by $\widetilde{X}, \widetilde{Y}, \widetilde{Z}$, along with Gauss law constraints as depicted in Figure 14. We modify the edge terms in the original Hamiltonian with $\widetilde{Z}$ on suitable faces to ensure their commutation with the Gauss law constraints. The local terms of the resulting Hamiltonian are depicted in Figure 15 after gauge fixing $Z^x = 1, Z^y = 1, Z^z = 1$, swapping the lattice with the dual lattice, and changing basis $X \leftrightarrow Z, \widetilde{X} \leftrightarrow \widetilde{Z}$.

**Fractionalization of the global relation.** The theory described directly above has planar subsystem symmetry given by the product $\prod X \prod \widetilde{Z}$ on an infinite plane, equivalent to a "dipole" of two parallel $X$ membranes enclosing a web of $\prod \widetilde{Z}$ on all edges of the plane.

The product of all planar subsystem symmetry on infinite planes in all basis directions results in the identity, since $X$ operators on edges cancel, and $\widetilde{Z}$ operators also cancel as every edge appears on two planes oriented in different directions. This is the global relation for planar subsystem symmetries of different directions.

On the other hand, on a finite-size region, the $X$ edges do not cancel in the product, but form a membrane of $X$s on the dual lattice on the boundary of the region. Thus the product does not commute with the Wilson line $\prod Z$ that pierces into the region. This implies that the global relation of the planar subsystem symmetry fractionalizes on the particle excitations associated to the Wilson line.

### 5.3 Fractionalization on loop excitations: Fractional charge under planar symmetry

Similar to the construction of fractionalization of ordinary symmetry in Refs. [2,3,21], here we realize fractionalization of planar subsystem one-form symmetry by using two-form symmetry. We achieve this by constructing a background for the two-form symmetry using the background foliated two-form gauge field for the planar subsystem symmetry:

$$B_3 = \sum_k \frac{q_k}{2} dB_2^k. \tag{52}$$

The background gauge field for this planar subsystem symmetry couples to a system with two-form symmetry as above. For odd $q_k$, this implies that the magnetic loop excitation carries a fractional charge under the subsystem planar symmetry on planes of constant $x^k$. For subsystem planar symmetry $G$, and two-form symmetry $\mathcal{A}^{(2)}$, such fractionalization classes belong to $H^3(B^2 G, \mathcal{A}^{(2)})$.

#### 5.3.1 Lattice Hamiltonian model

We now construct a lattice Hamiltonian model on the cubic lattice with the fractionalization described above.

Our starting point is the SSPT phase with $\mathbb{Z}_2$ ordinary symmetry and planar symmetry, described by the topological action with cocycle

$$\phi_4 = \pi a \cup B_3 = \pi a \cup \left( q^x B_2^x \cup_1 B_2^x + q^y B_2^y \cup_1 B_2^y + q^z B_2^z \cup_1 B_2^z \right), \tag{53}$$

where we have used $dB/2 = Sq^1 B = B \cup_1 B$ for any $\mathbb{Z}_2$ two-cocycle $B$. Next, we gauge the $\mathbb{Z}_2$ symmetry to obtain toric code enriched with the planar subsystem symmetry with loop fractionalization.

We introduce two qubits on each edge, one for the ordinary $\mathbb{Z}_2$ gauge field, the other for the planar subsystem symmetry on planes perpendicular to the edge. These two qubits are acted on by the Pauli operators $X, Y, Z$ and $X^k, Y^k, Z^k$, respectively, where $k = x, y, z$ labels the direction of the edge. We also let $a = (1-Z)/2$, $\lambda_1^k = (1-Z^k)/2$.

This leads to the following local commuting projector Hamiltonian for $\mathbb{Z}_2$ gauge theory enriched by planar subsystem symmetry with fractionalization on loop excitations:

$$H = \sum_v \left( \prod_{v \in \partial e} X_e \right) (-1)^{\int \widetilde{v} \cup \sum_k q^k d\lambda_1^k \cup_1 d\lambda_1^k} - \sum_f \prod_{e \in \partial f} Z_e - \sum_k \sum_{e_k} X_{e_k}^k (-1)^{\int a \cup \left( d\lambda_1^k \cup_2 d\widetilde{e}_k + \widetilde{e}_k \cup_1 d\widetilde{e}_k \right)}, \tag{54}$$

where $e_k$ are edges in the $x^k$ direction, the terms with $\prod Z_e$ are the flux terms, and the integrals in the integrals are over 3D space. Here, $\widetilde{v}$ and $\widetilde{e}$ are 0-cochain and one-cochain that takes value 1 on the vertex $v$ and edge $e$, while 0 when evaluated on other vertices and edges. For a review of higher cup products, see *e.g.* Ref. [2].

We remark that fractionalization on loop excitations for subsystem symmetry is also discussed in Ref. [25].

### 5.4   Fractionalization on loop excitations: Global relation for linear symmetry

Let us consider the fractionalization of two-form symmetry by its global relations. The $\mathbb{Z}_2$ gauge theory has Wilson line operators that generate two-form symmetry. We can consider three rigid subgroups generated by lines along the $x, y, z$ directions. The symmetry on the planes given by a product of lines enjoys foliation-independence, and this gives global relations between the linear symmetries. Such global relations are violated by the magnetic flux excitations that carry the holonomy of the Wilson line: small contractible loop operators equal $(-1)$ if the loop operators surround the magnetic flux. This is the direct analog to the fractionalization of global relations of one-form symmetry in 2+1D, where the generators are also one-dimensional. The difference is that such generators in 3+1D generate a two-form symmetry instead of a one-form symmetry. As in 2+1D, we can describe the fractionalization using different subgroup two-form symmetry embedded into the full two-form symmetry that transforms the magnetic membrane operator, where the background fields satisfy

$$B_3 = \sum_{k,l} B_3^{k,l}\,. \tag{55}$$

The fractionalization is similar to 2+1D case, and we do not repeat the discussion here.

### 5.5   Example of anomalous fractionalized subsystem symmetry

We can fractionalize subsystem one-form symmetry on both particle and loop excitations using

$$B_2 = \sum_k B_2^k\,, \qquad B_3 = \sum_k \frac{q_k}{2} dB_2^k\,. \tag{56}$$

This is because the one-form and two-form symmetries in lattice gauge theory already have a mixed anomaly, as described by the bulk SPT phase with the effective action

$$\frac{2}{2\pi} \int_{5d} B_2 B_3\,. \tag{57}$$

The subsystem one-form symmetry with the above fractionalization also has an anomaly. To see this, we substitute the above relations (56) into the effective action (57): this gives the bulk 4+1D SSPT phase with the effective action

$$\text{Bulk SSPT}: \quad \frac{2}{2\pi} \sum_{k,l} q_l \int_{5d} B_2^k dB_2^l\,. \tag{58}$$

## 6   Lattice models for linear subsystem symmetry fractionalization

In this section, we introduce lattice models with 2+1D topological order enriched by linear subsystem symmetry. The linear subsystem symmetries are fractionalized, which is described

by a group of Abelian anyons that decorate the corners of truncated subsystem symmetry operators. The first model produces linear subsystem symmetry-enriched topological (SSET) fractionalization corresponding to a group of Abelian anyons with bosonic mutual statistics in a 2+1D topological order with a gapped boundary to the vacuum. The second model produces linear SSET fractionalization corresponding to a group of arbitrary Abelian anyons at the 2+1D boundary of a 3+1D bulk. The topological order of the 2+1D boundary in this case need not admit a 1+1D gapped boundary to the vacuum and hence can be chiral. We conjecture that the SSET order of the 2+1D boundary in this model is anomalous whenever the Abelian anyons appearing in the fractionalization have a nontrivial $F$ symbol. This anomaly is labeled by an element of $H^3(G, U(1))$, where $G$ is the group of Abelian anyons involved in the fractionalization [61, 62].

The lattice models in this section are constructed from string nets based on fusion categories with an Abelian grading [63–68]. A fusion category [69] that is graded by a finite group $G$ decomposes into a direct sum of subcategories labeled by group elements

$$\mathcal{C}_G = \bigoplus_{g \in G} \mathcal{C}_g. \tag{59}$$

We use the notation $a_g$ for string type $a \in \mathcal{C}_g$. This means that the Hilbert space with basis states labeled by the finite set of string types (simple objects up to isomorphism), admits a natural $G$-grading

$$\mathbb{C}[\mathcal{C}_G] = \bigoplus_{g \in G} \mathbb{C}[\mathcal{C}_g], \tag{60}$$

with a basis for $\mathbb{C}[\mathcal{C}_g]$ given by $\big|a_g\big\rangle$, for all $a \in \mathcal{C}_g$. For an Abelian group $G$ this grading comes together with a diagonal unitary operator

$$\widehat{\chi}\big|a_g\big\rangle = \chi(g)\big|a_g\big\rangle, \tag{61}$$

where $\chi \in \widehat{G}$ is a character of $G$.

## 6.1  2+1D lattice model

Any 2+1D topological order $\mathcal{M}$ that admits a gapped boundary can be realized by a string-net model based on a fusion category $\mathcal{C}$, such that the center of $\mathcal{C}$ is $\mathcal{M}$ [56].[6] Furthermore any such topological order with a subgroup $\widehat{G}$ of mutually Abelian bosons can be realized by a string-net based on a $G$-graded fusion category $\mathcal{C}_G$. This follows by gauging the $G$ symmetry [4–6, 63, 71] of the symmetry-enriched string-nets introduced in Refs. [66–68].

The string-net model is defined on a directed honeycomb lattice with a $\mathbb{C}[\mathcal{C}]$ string degree of freedom on each edge [56]. Reversing the direction of an edge is equivalent to swapping each string type with its antistring $a \mapsto a^*$. We restrict our discussion to the multiplicity-free case for simplicity, lifting this restriction is straightforward and involves including an additional degeneracy space degree of freedom on every vertex and altering the vertex term in the Hamiltonian. The Hamiltonian governing the string-net model is

$$H_{\text{SN}} = -\sum_v A_v - \sum_p \sum_{a \in \mathcal{C}} \frac{d_s}{\mathcal{D}^2} B_p^a, \tag{62}$$

where $d_a$ is the quantum dimension of string type $a$, and $\mathcal{D}$ is the total quantum dimension

$$\mathcal{D}^2 = \sum_{a \in \mathcal{C}} d_c^2. \tag{63}$$

---

[6]Here $\mathcal{M} \cong \mathcal{Z}(\mathcal{C})$ denotes an anyon theory (modular tensor category), $\cong$ denotes braided equivalence of anyon theories, and $\mathcal{Z}$ denotes the Drinfeld center. See Ref. [70] for a review of modular tensor categories.

The vertex terms apply an energy penalty to states that do not obey the fusion rules of $\mathcal{C}$ at a vertex

$$A_v \quad \overset{i}{\underset{j}{\longrightarrow}} \overset{k}{} = N_{ij}^k \quad \overset{i}{\underset{j}{\longrightarrow}} \overset{k}{} \quad , \qquad A_v \quad \overset{j}{\underset{k}{\longrightarrow}} i = N_{i^*j^*}^{k^*} \quad \overset{j}{\underset{k}{\longrightarrow}} i \tag{64}$$

where $N_{ij}^k = \dim(\mathrm{Hom}(i \otimes j, k))$. The plaquette terms $B_p^a$ fluctuate between different string-net configurations by fusing a loop of string type $a$ into the lattice

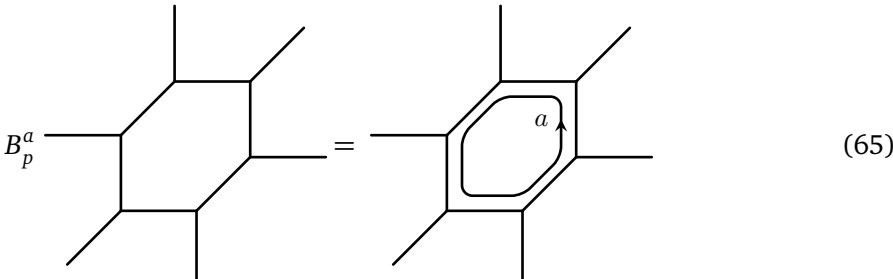

$$\tag{65}$$

see Ref. [56] for an explicit expression of the matrix elements that occur when such a loop is fused into the lattice. In the equation above we have assumed the vertex terms on the boundary of the plaquette are satisfied, outside of this subspace $B_p^a$ is defined to be 0. For the $B_p^a$ operator to act nontrivially, we further assume that the plaquette is punctured, in the sense that the $a$ loop cannot be removed via shrinking, but rather must be fused into the lattice.

In the string-net lattice models based on $\mathcal{C}_G$, the string operator that creates and moves a charge boson labeled by $\chi \in \widehat{G}$ is simply

$$S_{\widehat{\gamma}}(\chi) = \prod_{e \in \widehat{\gamma}} \widehat{\chi}^{\widehat{\gamma}(e)}, \tag{66}$$

where $\widehat{\gamma}$ is a path in the dual lattice, and $\widehat{\gamma}(e) = \pm 1$ for right and left handed crossings of $\widehat{\gamma}$ with $e$, respectively. Each $\chi$ boson appears as a plaquette excitation of the string-net model when the relevant plaquette operators $B_p^{a_g}$ take on the eigenvalue $d_{a_g}\chi^*(g)$.

We now make use of the simple structure of the bosonic string operators introduced in Eq. (66) to write down a modified string-net Hamiltonian with $\widehat{G}$ linear subsystem symmetry fractionalization associated to the $\chi \in \widehat{G}$ bosons. The model is defined by grouping the honeycomb lattice degrees of freedom of the string-net model into a square superlattice with two honeycomb vertices plus an additional $\mathbb{C}[G]$ degree of freedom per site. The Hamiltonian is

$$H_{\mathrm{LSS}} = -\sum_v \left( A_v^{(1)} + A_v^{(2)} + \frac{1}{|G|} \sum_{\chi \in \widehat{G}} (C_{v\widehat{x}}^\chi + C_{v\widehat{y}}^\chi) \right) - \sum_p \frac{1}{|G|} \sum_{g \in G} \sum_{a_g \in \mathcal{C}_g} \frac{d_{a_g}}{\mathcal{D}_1^2} B_p^{a_g}, \tag{67}$$

where $v$ runs over sites of the square superlattice and $\mathcal{D}_1$ is the total quantum dimension of the subcategory $\mathcal{C}_1$. This model includes the standard $A_v$ vertex terms of the string-net model shown in Eq. (64), there is a pair of such terms per vertex of the square superlattice. The additional $C_v^\chi$ vertex terms introduce a $\chi$ charge on the new degree of freedom at the vertex along with a quadruple of $\chi$ bosons, and their antiparticles,

$$C_{v\widehat{x}}^\chi \quad \overset{}{\longrightarrow}\bullet = \widehat{\chi}^\dagger \overset{}{\longrightarrow}\bullet_{\widehat{\chi}} \overset{\widehat{\chi}}{} \quad , \qquad C_{v\widehat{y}}^\chi \quad \overset{}{\longrightarrow}\bullet = \overset{\widehat{\chi}}{\longrightarrow}\bullet_{\widehat{\chi}^\dagger} \overset{}{\underset{\widehat{\chi}^\dagger}{}} \tag{68}$$

where $\widehat{\chi}|g\rangle = \chi(g)|g\rangle$ on the $\mathbb{C}[G]$ degree of freedom. We remark that the product

$$C_{v\widehat{x}}^{\chi} C_{v\widehat{y}}^{\chi} \qquad = \qquad$$ 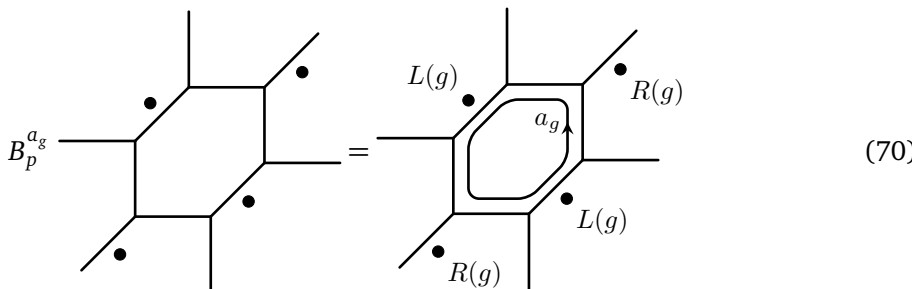 (69)

corresponds to a small loop, on the dual lattice around $v$, of the string operator for the $\chi$ boson which is not independent of $A_v^1$, $A_v^2$. Finally, the $B_p^{a_g}$ terms cause the string-net to fluctuate by fusing in a loop of string type $a_g$ while simultaneously applying a left, or right, $g$ multiplication to the adjacent vertices

$$B_p^{a_g} \qquad = \qquad$$ (70)

where $L(g)|h\rangle = |gh\rangle$ and $R(g)|h\rangle = |gh^{-1}\rangle$.

The $\widehat{G}$ linear subsystem symmetry of the above model is generated by

$$\prod_i \widehat{\chi}_{i,j}, \qquad \text{on rows and} \tag{71}$$

$$\prod_j \widehat{\chi}_{i,j}, \qquad \text{on columns.} \tag{72}$$

The $C_{v\widehat{x}}^{\chi}$, $C_{v\widehat{y}}^{\chi}$ terms in the Hamiltonian force the excitation created by applying a truncated linear subsystem symmetry generator labeled by $\chi$ to the ground state to simply create a particle-antiparticle pair of $\chi$, $\chi^*$ bosons at the truncated endpoint. One can easily verify that the product over all rows of the $\chi$ generators with the product over all columns of the $\chi^*$ generators is the identity operator, and hence there is a global $\widehat{G}$ relation. By construction, the domain wall obtained by truncating the global relation on a region is simply the string operator for the $\chi$ boson at the boundary of the region. Hence the symmetry fractionalization class is labeled by the $\chi$ boson [26].

## 6.2  3+1D lattice model

A 2+1D topological order described by an arbitrary anyon theory $\mathcal{M}$ can be realized at the boundary of a 3+1D Walker-Wang model based on $\mathcal{M}$ [72]. Each anyon theory admits a universal grading given by $G$ the dual of the group of all Abelian anyons therein $\widehat{G} \subseteq \mathcal{M}$ [73]. This grading is induced by the braiding phases of each anyon $a \in \mathcal{M}$ with all Abelian anyons $\chi \in \widehat{G}$

$$M_{a,\chi} = \chi(g), \tag{73}$$

for some $g \in G$ and all $\chi \in \widehat{G}$. Given Eq. (73) we assign $a \in \mathcal{M}_g$. This defines the universal $\widehat{G}$ grading of $\mathcal{M}$. The above braiding phases can be viewed as the charge of the string operator for anyon type $a$ under the 1-form symmetry generated by the loop operators of the Abelian

anyons $\widehat{G}$. We remark that this 1-form symmetry may be anomalous due to nontrivial $F$ and $R$ symbols on the group of Abelian anyons $\widehat{G}$ [74].

The Walker-Wang model is defined, similarly to the string-net model, on a resolved cubic lattice with a $\mathbb{C}[\mathcal{M}]$ anyon worldline degree of freedom on each edge.

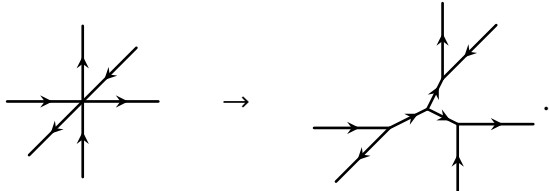

Again, reversing an edge is equivalent to exchanging the anyon type on that edge with its antiparticle. Here, we restrict our discussion to the case of a multiplicity-free input anyon theory for simplicity but the generalization is straightforward. The Hamiltonian is

$$H_{\text{WW}} = -\sum_v A_v - \sum_p \sum_{a\in\mathcal{M}} \frac{d_a}{\mathcal{D}^2} B_p^a, \tag{74}$$

where $d_a$ and $\mathcal{D}$ are the quantum dimension of $a$ and total quantum dimension respectively. The vertex term again applies an energy penalty to states that do not obey the fusion rules of $\mathcal{M}$

$$A_v \;\; = N_{i^*j^*}^{k^*} \;\; , \qquad A_v \;\; = N_{ij}^{k} \;\; , \tag{75}$$

$$A_v \;\; = N_{ij}^{k} \;\; , \qquad A_v \;\; = N_{i^*j^*}^{k^*} \;\; . \tag{76}$$

The plaquette terms fluctuate between different braided anyon worldline net configurations by fusing in a loop of each anyon type

$$B_p^a \;\; = \;\; , \qquad\qquad p \perp \widehat{x}, \tag{77}$$

$$B_p^a \;\; = \;\; , \qquad\qquad p \perp \widehat{y}, \tag{78}$$

$$B_p^a \;\; = \;\; , \qquad\qquad p \perp \widehat{z}. \tag{79}$$

We remark that the matrix elements of the plaquette operators above differ from the string-net models as the braiding of the anyon theory $\mathcal{M}$ has to be used to fuse the loop of $a$ into the lattice. See Ref. [72] for an explicit description of the matrix elements that result from the above operators.

For a modular anyon theory $\mathcal{M}$, there are no nontrivial point excitations in the bulk of the Walker-Wang model as all string operators are confined by the $B_p^a$ terms they pass through due to braiding nondegeneracy. If the Walker-Wang model is terminated on a smooth surface, with no dangling edges, the string operators corresponding to anyons $a \in \mathcal{M}$ are no longer confined when applied to the smooth exterior of the surface. Hence the surface of the Walker-Wang model based on a modular anyon theory $\mathcal{M}$ hosts a topological order described by $\mathcal{M}$, while the bulk lies in the trivial phase [72].[7]

In the Walker-Wang model based on $\mathcal{M}$ the operator $\widehat{\chi}$ applied to an edge $e$ can be viewed as creating a small loop of Abelian anyon type $\chi \in \widehat{G}$ encircling $e$, following Eq. (73).[8] A loop $\widehat{\gamma}$ of $\chi$ anyon type in the dual lattice creates an excitation of the model, with eigenvalue $d_{a_g} \chi^*(g)$ under the plaquette operators $B_p^{a_g}$ that $\widehat{\gamma}$ passes through. A product of $\widehat{\chi}$ operators over a surface $\widehat{\varsigma}$ in the dual lattice

$$\prod_{e \in \widehat{\varsigma}} \widehat{\chi}_e, \tag{80}$$

creates a loop of anyon type $\chi$ at the boundary of the surface $\widehat{\partial \varsigma}$. Hence such products over closed surfaces $\widehat{\varsigma}$ in the dual lattice generate a $\widehat{G}$ 1-form symmetry of the model as they commute with the Hamiltonian and create no excitations. The product of $\widehat{\chi}$ operators over a surface $\widehat{\varsigma}$ that ends on a smooth boundary $S$, i.e. $\widehat{\delta \varsigma} \subset S$, creates a loop operator for the anyon type $\chi \in \widehat{G}$ which is deconfined on the boundary following the discussion above.

We now combine the symmetry structure of the Abelian anyon string types with the boundary properties of the Walker-Wang model to define a new model with $\widehat{G}$ linear subsystem symmetry fractionalization in the bulk that is associated to $\chi \in \widehat{G}$ anyons on the boundary. The model is defined by adding two $\mathbb{C}[G]$ degrees of freedom to each site of the Walker-Wang model on a cubic lattice. The Hamiltonian is

$$H_{\text{ALSS}} = -\sum_v \left( A_v^1 + A_v^2 + A_v^3 + A_v^4 + \frac{1}{|G|} \sum_{\chi \in \widehat{G}} (C_{v\widehat{x}}^\chi + C_{v\widehat{y}}^\chi + C_{v\widehat{z}}^\chi) \right) - \sum_p \frac{1}{|G|} \sum_{g \in G} \sum_{a_g \in \mathcal{C}_g} \frac{d_{a_g}}{\mathcal{D}_1^2} B_p^{a_g}, \tag{81}$$

where $v$ runs over sites of the cubic lattice, and $\mathcal{D}_1$ is the total quantum dimension of the subcategory $\mathcal{M}_1$. This model includes the four types of $A_v$ vertex term from the Walker-Wang model in Eq. (75). There are additional $C_v^\chi$ vertex terms that cause composites of $\chi$ charges on the site degrees of freedom and plaquette excitations, corresponding to small loops of $\chi$ anyon string, to fluctuate

$$C_{v\widehat{x}}^\chi \quad \underset{\text{[diagram]}}{\phantom{X}} = \quad \underset{\text{[diagram]}}{\phantom{X}}, \tag{82}$$

[7]An interesting subtlety arises when trivializing the bulk of a chiral Walker-Wang model as it seems to require a nontrivial quantum cellular automaton [75–77].

[8]$\widehat{\chi}$ defines a $\widehat{G}$-grading on the edges of the Walker-Wang model, and similar to a graded string-net it can be constructed by gauging a 1-form symmetry-enriched Walker-Wang model [64, 65].

$$C_{v\widehat{y}}^{\chi} \quad = \quad \hat{\chi}^{\dagger} \quad \hat{\chi} \quad , \tag{83}$$

$$C_{v\widehat{z}}^{\chi} \quad = \quad \hat{\chi}^{\dagger}\hat{\chi}^{\dagger} \quad , \tag{84}$$

where $\widehat{\chi}|g\rangle = \chi(g)|g\rangle$ on the $\mathbb{C}[G]$ degree of freedom. Similar to the 2+1D model, the product

$$C_{v\widehat{x}}^{\chi}C_{v\widehat{y}}^{\chi}C_{v\widehat{z}}^{\chi} \quad = \quad \hat{\chi}^{\dagger} \quad \hat{\chi}^{\dagger} \quad \hat{\chi} \quad \tag{85}$$

corresponds to the small worldsheet of a $\chi$ boson string operator, in the dual lattice around $v$, this term is not independent of $A_v^1, A_v^2, A_v^3, A_v^4$. Finally, the $B_p^{a_g}$ terms again cause the string-net to fluctuate by fusing in a loop of string type $a_g$ while simultaneously applying a left, or right, $g$ multiplication to the adjacent vertices

$$B_p^{a_g} \quad = \quad , \qquad p \perp \widehat{x}, \tag{86}$$

$$B_p^{a_g} \quad = \quad , \qquad p \perp \widehat{y}, \tag{87}$$

$$B_p^{a_g} \quad = \quad , \qquad p \perp \widehat{z}, \tag{88}$$

where $L(g)|h\rangle = |gh\rangle$ and $R(g)|h\rangle = |gh^{-1}\rangle$.

The model introduced in Eq. (81) has a $\widehat{G}$ linear subsystem symmetry generated by

$$\prod_i \widehat{\chi}_{i,j,k}, \qquad \text{along } \widehat{x}, \tag{89}$$

$$\prod_j \widehat{\chi}'_{i,j,k}, \qquad \text{along } \widehat{y}, \tag{90}$$

$$\prod_k \widehat{\chi}\,\widehat{\chi}'_{i,j,k}, \qquad \text{along } \widehat{z}, \tag{91}$$

directions of the lattice, respectively. The $C^\chi_{v\widehat{x}}$, $C^\chi_{v\widehat{y}}$, $C^\chi_{v\widehat{z}}$ terms in the Hamiltonian (81) enforce that the excitation created by acting on the ground state with a truncated linear subsystem symmetry generator, labeled by the element $\chi$, is equivalent to a loop of $\chi$ anyon string encircling the edge adjacent to the truncation. Once again it is easy to verify that the product over all linear $\chi$ generators along the $\widehat{x}$ and $\widehat{y}$ direction with the product over all $\chi^*$ generators along the $\widehat{z}$ direction gives the identity operator, and hence there is a global $\widehat{G}$ relation. By construction, the domain wall obtained by truncating the global relation on a region with a nonempty surface is simply the 1-form operator that corresponds to sweeping a string operator for the $\chi$ anyon over the domain wall surface.

To introduce a smooth boundary along a lattice plane to the above model we simply remove dangling legs from the relevant terms in the Hamiltonian (81), while retaining all site degrees of freedom. The $\widehat{G}$ symmetry and its global relation persist in the presence of such a boundary. The truncated global relation on a region that ends on the boundary results in a domain wall that also ends on the boundary. The action of this domain wall is equivalent to applying a loop of $\chi$ boson to the boundary theory. Hence the symmetry fractionalization class of the boundary theory is labeled by the $\chi$ boson [26]. Acting on the ground space of the model we have

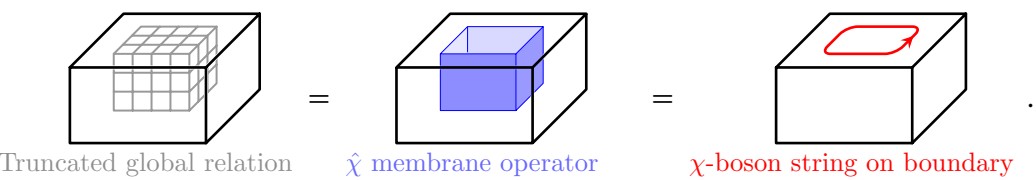

$$\underbrace{\hspace{5em}}_{\text{Truncated global relation}} = \underbrace{\hspace{5em}}_{\widehat{\chi} \text{ membrane operator}} = \underbrace{\hspace{5em}}_{\chi\text{-boson string on boundary}}.$$

# 7 Discussion

In this work, we have introduced a number of field theories and lattice models with topological order that exhibit subsystem symmetry fractionalization. Our approach was based on a general method we developed to derive subsystem symmetry from higher-form symmetry, which applies to both lattice models and field theories. With this method, we were able to find models that exhibit anomalous subsystem symmetry fractionalization, which requires either a higher dimensional bulk or a non-onsite symmetry to implement. Our models include subsystem symmetry-enriched $\mathbb{Z}_2$ lattice gauge theories in 2+1D and 3+1D, as well as string-net and Walker-Wang models, which can support nonabelian anyons. The method we have developed here extends the theory of subsystem symmetry fractionalization and deepens the existing connection with the well-developed subject of anyon theories.

The progress reported here raises a number of questions about the phases, phase transitions and applications of SSETs, which we leave to future work. We have organized these questions according to topic below.



**Complete classification of SSETs**

- Do the existing models of subsystem symmetry fractionalization in 2+1D and 3+1D exhaust all possible phenomena? Alternatively, there may exist new forms of subsystem symmetry fractionalization that are yet undiscovered.

- Is there a distinction between strong and weak SSET phases [31, 78]?

- Can SSETs be classified via bifurcating fixed points under symmetry-respecting entanglement renormalization [79]?

- Is there a topological defect network construction of all SSET phases [80–82]?

- Can we understand subsystem symmetry fractionalization as being inherited from conventional symmetry fractionalization on sub-dimensional systems via a coupled layer construction [44, 83]?

- Can all SSETs be obtained by starting with an appropriate SSPT and gauging a subgroup of the full symmetry group [25, 26]?

- It was shown in Ref. [25] that subsystem symmetry-enrichment can lead to spurious contributions to the topological entanglement entropy. Can the subsystem symmetry-enrichment class be detected via such features in the entanglement entropy [84–87]?

- Can fractal SSETs be consructed and classified within the current framework [26, 32, 88, 89]?

**Phase transitions protected by SSET order**

- What kind of phase transitions occur due to the breaking of subsystem symmetry in an SSET [44]?

- What kind of boundaries can occur at the edge of an SSET [90–92]? We rematk that the boundary can have non-invertible symmetry described by the symmetry TQFT of the SSET, see e.g. Refs. [93–95].

**Application to quantum computation**

- Generalized symmetries have proven fruitful for discovering new fault-tolerant logical gates, see e.g. Refs. [96–99]. Do the new forms of subsystem symmetry-enrichment introduced here lead to new logical gates?

- Are SSETs useful resources for universal measurement-based quantum computation, similar to SSPTs [29, 100–102]?

# Acknowledgments

We thank José Garre-Rubio and Michael Hermele for collaboration during the early stages of this work. We thank Meng Cheng and Ho Tat Lam for useful discussions and comments on a draft. We thank Xie Chen and Kevin Slagle for comments on a draft.

**Funding information** The work of P.-S.H. is supported by the Simons Collaboration on Global Categorical Symmetries. DTS and AD are supported by the Simons Collaboration on Ultra-Quantum Matter, which is a grant from the Simons Foundation (DTS: 651440, AD: 651438). The work of DW on this project was supported by the Australian Research Council Discovery Early Career Research Award (DE220100625).

# A  Foliated gauge fields

In this appendix, we summarize some properties of foliated gauge fields, following the conventions in Ref. [39].

We denote the foliation one-form by $e^k$ with $k$ labeling the foliations, which satisfies $e^k e^k = 0$. For instance, in flat Euclidean spacetime with coordinates $(t, \{x^i\})$ we take $e^k = dx^k$,

## A.1  One-foliated gauge fields

Let us denote an $n$-form gauge field $B_n^k$ with $k$ labelling foliations, it must satisfy

$$B_n^k e^k = 0, \tag{A.1}$$

where $k$ is not summer over, and $e^k$ are foliation one-forms (in this work we always take them to be $dx^k$ for space coordinates $x^k$). The above field has gauge transformation $B_n^k \to B_n^k + d\lambda_{n-1}^k$ for an $(n-1)$-form $\lambda_{n-1}^k$ that satisfies $\lambda_{n-1}^k e^k = 0$.

We use $A_n^k$ to denote an $n$-form gauge field with the gauge transformation

$$A_n^k \to A_n^k + \alpha_n^k + d\lambda_{n-1}^k, \tag{A.2}$$

where $\alpha_n^k e^k = 0$.

We refer to the above gauge fields $A_n^k, B_n^k$ as 1-foliated gauge fields since they satisfy constraints involving only one foliation one-form. They naturally describe gauge fields living on codimension-one leaves of a manifold.

## A.2  Higher-foliated gauge fields

We define 2-foliated gauge fields as follows: denote an $n$-form 2-foliated gauge field $B_n^{k,l}$ which must satisfy

$$B_n^{k,l} e^k e^l = 0, \tag{A.3}$$

where $k, l$ are not summed over. Similarly, $A_n^{k,l}$ has addition gauge transformations by shifting with any $\alpha_n^{k,l}$ that satisfies $\alpha_n^{k,l} e^k e^l = 0$. Such 2-foliated gauge fields naturally live on the codimension-two intersection of codimension-one leaves. It is straightforward to generalize this recipe to higher-foliated gauge fields.

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
