# Peer review of "Subsystem Symmetry Fractionalization and Foliated Field Theory"

_SciPost Physics, doi:SciPost Phys. 18, 147 (2025)_

## Round 1 · Referee Report · Anonymous (Referee 1) · 2024-5-30

Report

This work explores enrichment of topological order with subsystem symmetries in various dimensions. This is done by embedding subsystem symmetries into higher-form symmetries as rigid subgroups. The general constructions are demonstrated neatly in several examples with explicit Hamiltonians.

While the results are very interesting and are worth publishing in this journal, I have some concerns/questions listed below:

  1. My main concern is in the section on "gauging contractible one-form symmetry". In the example discussed by the authors, the starting point is the 2+1d toric code with a vertex and a plaquette term, whereas the resulting model has only the plaquette term (on the dual lattice). While the 1-form symmetry is topological in the new model, its spectrum contains a lot more ground states (because of the absence of vertex terms in the Hamiltonian) than the toric code. In other words, the new model is not described by the $\mathbb Z_2$ gauge theory at low energies. In what sense is this topological 1-form symmetry of the new model related to the topological 1-form symmetry in field theory?

  2. I am puzzled by the operators in the lower part of fig 2. The untruncated versions of these operators are trivial because they can be written as product of Gauss law operators in fig 1, which are imposed exactly. The nontrivial 1-form symmetry operators are still $\prod X$ over curves on dual lattice, which are now fully topological. So, it's not clear to me what the authors mean by "[the operators in the lower part of fig 2] satisfy a global relation" when they are in fact trivial.

  3. Related to the point above, the authors say that the edge-qubits can be removed to get the new model with only face-qubits. In ordinary gauging, this works fine because the Gauss law involves only one $X$ so the edge-qubits can all be gauge-fixed. On the other hand, the Gauss laws in fig 1 involve two $X$'s each, so gauge-fixing needs more care. In fact, on a finite lattice with periodic b.c., I do not believe they can all be gauged-fixed because of the nontrivial 1-form symmetry operators mentioned in the previous point. This should be clarified.

While the above concerns should be addressed, I believe the main results of the paper are not significantly affected by them. There are also some minor issues listed below:

  1. On page 7, I think $(n-1)$ and $n$ should be exchanged in the sentence: "Furthermore, since the fully topological $(n-1)$-form symmetry on any contractible submanifolds $\delta$ is trivial, the new $n$-form symmetry on such a submanifold $\Sigma$ does not depend on the foliation."

  2. Below eq (2.4), the authors write: "The quotient is the product of contractible one-form symmetries." It is not clear to me why this is the case. For example, I don't think the line operator $Q^x$ at a fixed $x$ is contractible. Can the authors please explain?

  3. In eq (2.6), why does $\lambda^k$ not depend on $x^k$? I would assume that the gauge fields $A^k$ at different layers labelled by $x^k$ have their own gauge parameters, so I would expect that $\lambda^k$ depends on $x^k$ too.

  4. There are several typos: For example, in eqs (6.19)-(6.21), the superscript $a$ on $B_p$ is missing. Also, in the first paragraph of sec 5.4, there is a broken sentence.

Recommendation

Ask for minor revision

---

## Round 1 · Referee Report · Anonymous (Referee 2) · 2024-6-17

Report

In this paper, which appears to be a follow up of references [22] and [23], the authors study subsystem symmetries which naturally arise in systems with ordinary higher-form symmetries. Their analysis occurs both on the lattice and in the continuum, and in the latter context they use a foliated field theory approach. The continuum perspective is one of the ways this paper distinguishes itself from its antecedents.

One of the main examples is a 2+1d topological order with 1-form symmetry: the 1-dimensional symmetry operators of such a system, when inserted on lines which foliate space, can be thought of as furnishing a linear subsystem symmetry. They analyze the possible patterns of fractionalization of the “global relation” enjoyed by models with subsystem symmetries. One of the reasons this is interesting is because the global relation appears to be one of the main features which distinguishes a non-trivial subsystem-symmetric phase from one which is obtained by trivially layering lower-dimensional systems. The continuum perspective taken in many parts of this paper will hopefully make fractons and fracton-adjacent physics more accessible to those in the high energy community who think about generalized global symmetries. I recommend this paper for publication.

I offer a few comments and ask a few questions below.

The use of the terminology “subsystem one-form symmetry” (e.g. around equation 1.2) is somewhat confusing on first read. The authors seem to mean “0-form subsystem symmetry which is a subgroup of an ordinary one-form symmetry” but the language used might leave readers confusing it with what is sometimes called a “one-form subsystem symmetry” which is, for example, the kind of symmetry that the X-cube model has. I think the authors can be more careful and consistent with their language throughout.

This paragraph is regarding the toric code example analyzed in Section 1.1.1. The standard gauging of the $\mathbb{Z}_2$ one-form symmetry of the 2+1d toric code looks very different from what is carried out in the text. For example, in the standard approach, the end result of the gauging is essentially the 2+1d transverse field Ising model on the dual lattice. The difference between these two gauging procedures appears to stem from the Gauss law terms used in Figure 1, which differ from the more conventional choice $\tilde{X}_p X_e \tilde{X}_{p’}$, where $e$ is an edge and $p$ and $p’$ are the two plaquettes which share $e$ on their boundaries. Can the authors comment on the relationship between their approach and the more standard approach? Perhaps the difference is due to the fact that the authors say they are gauging the “contractible part of the one-form symmetry”, but it is not entirely clear to me what gauging a contractible one-form symmetry means in general.

Some small typos/comments:
-- Incomplete sentence on page 4: “Starting from a model where the higher-form symmetry is only topological on the states without higher-form charge.”
-- Around equation 1.2, three sentences start with “For instance, …”
-- “Systems with fully mobile excitations generically possess n-form higher symmetries that act on the extended operators that create these excitations [10].” I don’t know that I’d use the word “generically” to describe systems with a symmetry. Perhaps “often”?
-- There should not be a period in equation 2.10.
-- On page 13: “subsystem system symmetries”

Recommendation

Ask for minor revision

---

## Round 2 · Referee Report · Anonymous (Referee 2) · 2025-4-11

Report

I'm happy with the changes made and recommend the paper for publication.

Recommendation

Publish (meets expectations and criteria for this Journal)

---

## Round 2 · Referee Report · Anonymous (Referee 1) · 2025-4-14

Report

The revised manuscript addresses the concerns raised by the referees.

Recommendation

Publish (meets expectations and criteria for this Journal)

---

## Round 2 · Author Response

We thank the Referees for their detailed reports, we have responded to their questions and comments below. An updated manuscript with the changes outlined has been resubmitted. With these changes, we believe the updated draft is ready for publication.

Referee 1

We respond to the referee’s comments in order below:

  1. We have added the following paragraph to clarify this point:

“Since the original one-form symmetry on non-contractible cycles is still nontrivial after ``gauging the contractible one-form symmetry'', the resulting theory still has anomalous $\mathbb{Z}_2\times\mathbb{Z}_2$ one-form symmetry, which guarantees nontrivial ground state degeneracy.”

  1. We have added the following paragraph to clarify that due to gauging a subsystem subgroup of the 1-form symmetry of toric code, the line operators referenced do not become trivial, and so can satisfy a nontrivial global relation:

“The Gauss law in Figure 2 is different from the usual Gauss law imposed when gauging ordinary one-form symmetry (see e.g.~Refs). In the ordinary case, the Gauss law term is $\hat X_pX_e\hat X_{p'}$ for every edge $e$, and the two adjacent plaquettes $p,p'$. Such a Gauss law constraint is stronger than the one imposed in Figure 1. The latter constraint does not imply that all closed $X$-loops become trivial, while the former does. The $X$-loops that become trivial under the former Gauss law constraints are precisely those that are contractible, i.e. formed by product of the Hamiltonian star terms.”

  1. We have added a comment to clarify the fixing of edge degrees of freedom:

“On an infinite planar lattice, using the Gauss law constraints we can project out and remove the edge qubits, to arrive at a new model with face qubits only.\footnote{ On a more general lattice, we cannot gauge-fix the edge qubit completely.}”

  1. We thank the referee for pointing out this typo, we have corrected it in the updated submission.

  2. Below eq. 2.4 we have added a comment to clarify this point:

“The quotient is generated by the symmetry with current \begin{equation} J_0:=j_0^x-j_0^y,\quad J_x:=j_x^y,\quad J_y:=-j_y^x,\quad \partial_0J_0+\partial_xJ_x+\partial_yJ_y=0~. \end{equation} In other words, by gauging the symmetry we impose the global relation. We remark that since the relation demands that product of lines in $x$ directions to be equal to product of lines in $y$ directions, while individually a line is not contractable, the product of suitable $x,y$ lines are contractible, and they can be expressed as product of small contractible loops. “

  1. Below eq. 2.6 we have added clarification:

“where $\lambda^x=\lambda^x(t,y)$ and $\lambda^y=\lambda^y(t,x)$ to the maintain the vanishing components $A^x_x=0$ and $A^y_y=0$ that do not couple to the current.”

  1. We thank the referee for pointing out these typos, they have been corrected in the updated submission.

Referee 2

We respond to the referee’s comments below.

The use of the terminology “subsystem one-form symmetry” (e.g. around equation 1.2) is somewhat confusing on first read. The authors seem to mean “0-form subsystem symmetry which is a subgroup of an ordinary one-form symmetry” but the language used might leave readers confusing it with what is sometimes called a “one-form subsystem symmetry” which is, for example, the kind of symmetry that the X-cube model has. I think the authors can be more careful and consistent with their language throughout.

We have added the following paragraph above the first use of “subsystem one-form symmetry” to clarify the terminology:

“We use the following terminology throughout this work: a global symmetry, as characterized by a generator that commutes with the Hamiltonian, is called a $q$-form symmetry whenever the generator has support on a codimension-$q$ subspace in space. A symmetry is called a subsystem symmetry if the symmetry generator is not fully topological, i.e. the eigenvalue of the generator changes if we deform its support in general directions, even when it is away from other operators. These adjectives apply to global symmetries, and when a symmetry obeys the above two properties, we call it a subsystem $q$-form symmetry.”

This paragraph is regarding the toric code example analyzed in Section 1.1.1. The standard gauging of the Z2 one-form symmetry of the 2+1d toric code looks very different from what is carried out in the text. For example, in the standard approach, the end result of the gauging is essentially the 2+1d transverse field Ising model on the dual lattice. The difference between these two gauging procedures appears to stem from the Gauss law terms used in Figure 1, which differ from the more conventional choice XpXeXp′, where e is an edge and p and p′ are the two plaquettes which share e on their boundaries. Can the authors comment on the relationship between their approach and the more standard approach? Perhaps the difference is due to the fact that the authors say they are gauging the “contractible part of the one-form symmetry”, but it is not entirely clear to me what gauging a contractible one-form symmetry means in general.

We have added a paragraph to clarify our procedure to gauge the subsystem symmetry subgroup of the one-form symmetry of the 2+1d toric code:

“The Gauss law in Figure 2 is different from the usual Gauss law imposed when gauging ordinary one-form symmetry (see e.g.~Refs). In the ordinary case, the Gauss law term is $\hat X_pX_e\hat X_{p'}$ for every edge $e$, and the two adjacent plaquettes $p,p'$. Such a Gauss law constraint is stronger than the one imposed in Figure 1. The latter constraint does not imply that all closed $X$-loops become trivial, while the former does. The $X$-loops that become trivial under the former Gauss law constraints are precisely those that are contractible, i.e. formed by product of the Hamiltonian star terms.”

Some small typos/comments

We thank the referee for pointing out these typos, they have been corrected in the updated submission.

---

## Editorial Decision

published